# Offshore low-level jet observations and model representation using lidar buoy data off the California coast

Lindsay M. Sheridan[1], Raghavendra Krishnamurthy[1], William I. Gustafson Jr[1], Ye Liu[1],
Brian J. Gaudet[1], Nicola Bodini[2], Rob K. Newsom[1], and Mikhail Pekour[1]

[1]Pacific Northwest National Laboratory, Richland, Washington, United States
[2]National Renewable Energy Laboratory, Golden, Colorado, United States

*Correspondence to*: Lindsay M. Sheridan (lindsay.sheridan@pnnl.gov)

**Abstract.** Low-level jets (LLJs) occur under a variety of atmospheric conditions and influence the available wind resource for wind energy projects. In 2020, lidar-mounted buoys owned by the U.S. Department of Energy (DOE) were deployed off the California coast in two wind energy lease areas administered by the Bureau of Ocean Energy Management: Humboldt and Morro Bay. The wind profile observations from the lidars and collocated near-surface meteorological stations (4 m – 240 m) provide valuable year-long analyses of offshore LLJ characteristics at heights relevant to wind turbines. At Humboldt, LLJs were associated with flow reversals and north-northeasterly winds, directions that are more aligned with terrain influences than the predominant northerly flow. At Morro Bay, coastal LLJs were observed primarily during northerly flow as opposed to the predominant north-northwesterly flow. LLJs were observed more frequently in colder seasons within the lowest 250 m above sea level, in contrast with the summertime occurrence of the higher altitude California coastal jet influenced by the North Pacific High which typically occurs at heights of 300 m – 400 m.

The lidar buoy observations also validate LLJ representation in atmospheric models that estimate potential energy yield of offshore wind farms. The European Centre for Medium-Range Weather Forecasts Reanalysis version 5 (ERA5) was unsuccessful at identifying all observed LLJs at both buoy locations within the lowest 200 m. An extension of the National Renewable Energy Laboratory (NREL) 20-year wind resource dataset for the Outer Continental Shelf off the coast of California (CA20-Ext) yielded marginally greater captures of observed LLJs using the Mellor–Yamada–Nakanishi–Niino (MYNN) planetary boundary layer (PBL) scheme than the 2023 National Offshore Wind data set (NOW-23) which uses the Yonsei University (YSU) scheme. However, CA20-Ext also produced the most LLJ false alarms, instances when a model identified an LLJ but no LLJ was observed. CA20-Ext and NOW-23 exhibited a tendency to overestimate the duration of LLJ events and underestimate LLJ core heights.

## 1 Introduction

Offshore wind in the United States is in an early yet enthusiastic phase as coastal states begin to adopt it as a solution to meet local and national renewable energy goals. While the initial offshore wind development push has targeted the U.S. Atlantic coast, the U.S. Pacific coast is next in the queue with a development and operational pipeline of over 6,000 MW of potential

offshore wind generating capacity (Musial et al., 2023). Five wind energy lease areas in California, two off of Humboldt County and three off of Morro Bay, were in effect as of January 2024 (BOEM, 2024). In 2020, two DOE research buoys mounted with lidars and near-surface meteorological and oceanographic (metocean) instrumentation were deployed at the Humboldt and Morro Bay wind energy lease areas (Figure 1) to provide year-long observations of the wind resource at
heights relevant to offshore wind turbines (Krishnamurthy et al., 2023).

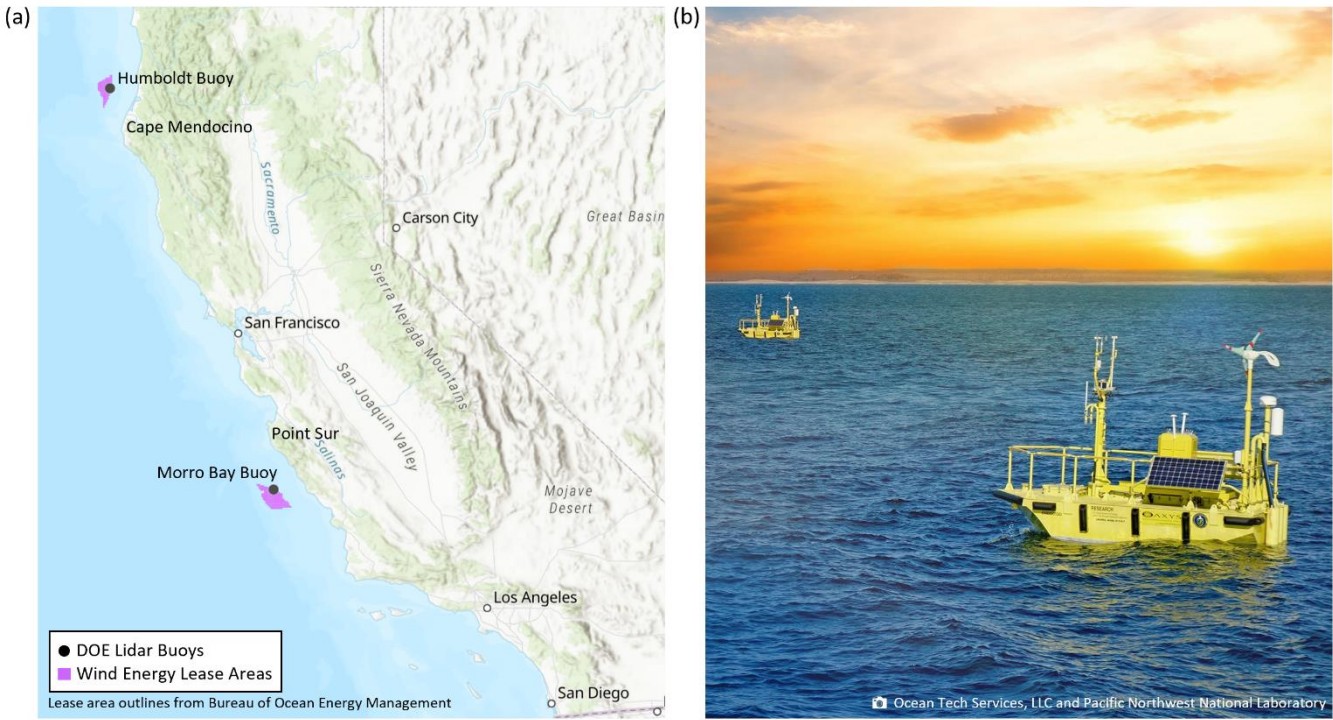

**Figure 1.** (a) Map of locations of the California DOE lidar buoy deployments and the California wind energy lease areas as of January 2024. (b) Photo of the DOE lidar buoys by Ocean Tech Services, LLC and Pacific Northwest National Laboratory.

A variety of meteorological phenomena are known to impact onshore and offshore wind profiles and therefore potential
offshore wind energy production, including frontal passages, downslope winds, sea breezes, and LLJs (Sherry and Rival, 2015; Nunalee and Basu, 2013; Aird et al., 2022; Liu et al., 2023; McCabe and Freedman, 2023). The last of these phenomena, LLJs, are wind speed maxima that occur in the lowest atmospheric altitudes and impact wind energy production in numerous ways. LLJs can result in significant acceleration of the wind speed at heights within the wind turbine rotor-swept area (Banta et al., 2008), which can lead to increases in wind energy generation. Gadde and Stevens (2021) noted the
importance of the placement of the LLJ in wind turbine wake recovery, with wakes recovering rapidly when LLJs are above the rotor-swept area due to enhanced turbulent mixing in the positive shear region below the LLJ. In the south-central Plains of the United States, Wimhurst and Greene (2019) determined an inverse relationship between the frequencies of LLJs and wind energy ramp events. From an engineering viewpoint, Gutierrez et al. (2016) reported increases in wind turbine fatigue during LLJ events due to sustained high energy and wind shear.

The summertime California coastal LLJ is well-studied and occurs due to the pressure gradient between the North Pacific High and southwestern U.S. thermal low (Burk and Thompson, 1996; Holt, 1996; Parish 2000; Pomeroy and Parish, 2001; Ström and Tjernström, 2004; Liu et al., 2023). The jet occurs at the top of the marine boundary layer with frequent core heights of 300 m – 400 m (Parish 2000), which are above the 2022 average offshore wind turbine hub height (116.6 m) and rotor diameter (174.6 m) (Musial et al., 2023). However, Musial et al. (2023) reported significantly larger hub heights (near 160 m) and rotor diameters (near 250 m) from future project announcements, leading to prospective rotor-swept upper limits approaching summertime LLJ core heights. In addition to the summertime California coastal LLJ, offshore LLJs can occur due to a variety of other mechanisms, including land-water temperature contrasts resulting in a thermal wind, topographic forcing due to inland elevation changes or complex coastal features such as capes and peninsulas, upwelling, and frontal passages (Parish, 2000).

Since offshore wind profile observations are sparse in temporal and geographic coverage, analysts often rely on models, such as reanalyses, and dedicated wind resource datasets to generate wind energy production estimates. ERA5 is an especially utilized reanalysis for wind resource assessment as both a standalone product and as a boundary condition input for higher-resolution models (Gevorgyan, 2018; Olauson, 2018; Dörenkämper et al., 2020; Soares et al., 2020; Hayes et al., 2021; de Assis Tavares et al., 2022). Given the importance of LLJs on wind energy production and structural integrity, it is essential to explore the degree to which model datasets represent LLJs. At four locations in the Baltic Sea, Hallgren et al. (2020) found that ERA5 and two regional reanalyses vastly underestimated the frequency of observed offshore LLJs and suspected the cause to be excessive turbulent mixing in the models during stable conditions. Similarly, Kalverla et al. (2019) found ERA5 unable to reliably reproduce offshore LLJs in the North Sea, with the model jets appearing higher and weaker than observed jets. One model component that influences the accuracy of simulated LLJs, and wind resource assessments in general, is the selection of PBL scheme. Nunalee and Basu (2014) performed sensitivity studies using the Weather Research and Forecasting (WRF) model with six unique boundary layer parameterizations to simulate offshore LLJs in the U.S. mid-Atlantic region and found that while all PBL schemes struggled to represent the intensity and structure of observed LLJs, the degree of accuracy was variable. Gevorgyan (2018) assessed WRF-simulated LLJs over Yerevan, Armenia, where observed LLJs are associated with local terrain influences, using nine distinct PBL schemes and found significant variation among the resultant jet core speeds. Svensson et al. (2016) found that no single PBL scheme among six evaluated in WRF outperformed the others in representing the wind speed and temperature profiles, the jet core height and wind speed, and the maximum wind shear during three LLJ case studies over the Baltic Sea.

The objectives of this study are to characterize offshore LLJs and the associated metocean conditions off the California coast using observations and to subsequently validate the performance of wind resource numerical datasets and reanalyses in LLJ representation. A discussion of the wind and metocean observations collected during the California deployments of the DOE lidar buoys is provided in Sect. 2, along with descriptions of the models compared in the validation portion of this study. Sect. 3 provides an analysis of the characteristics of observed offshore LLJs, including frequency, duration, temporal trends, and terrain influences. Sect. 3 concludes with a discussion of the relationships between LLJs and meteorological

conditions below the jet core. Sect. 4 examines the performance of models, including the global reanalysis ERA5 and two NREL-produced regional offshore wind resource datasets that employ distinct PBL schemes, in representing the occurrence of LLJs, along with the how accurately they capture features of observed LLJs, including core height, core speed, and duration. Finally, Sect. 5 summarizes the findings to evaluate the potential impact of offshore LLJs on future wind farms in the California offshore wind energy lease areas, along with information on the capabilities and limitations of LLJ representation in wind resource models.

## 2 Data discussion and methodology

Pacific Northwest National Laboratory manages multiple AXYS WindSentinel™ research buoys for DOE. The buoys are mounted with Leosphere WindCube 866 lidar systems and surface meteorological and oceanographic instrumentation (Severy et al., 2021). In advance of their deployment off the California coast, two of the DOE lidar buoys were validated by Ocean Tech Services and Det Norske Veritas at Woods Hole Oceanographic Institute's Martha's Vineyard Coastal Observatory. An International Electrotechnical Commission-certified reference lidar approximately 250 m away on an offshore platform was employed to validate the lidar buoys. The comparison yielded wind speed coefficients of determination ($R^2$) exceeding 0.98 and wind direction $R^2$ values exceeding 0.97 at heights up to 200 m above sea level (a.s.l.) (Gorton and Shaw, 2020).

The buoys were deployed off the northern and central coasts of California in the fall of 2020 (Figure 1). The central buoy was deployed from 29 September 2020 to 19 October 2021 in a depth of 1100 m of water approximately 40 km from the coast near Morro Bay (35.7107°N, 121.8581°W). The northern buoy was deployed from 8 October 2020 to 28 June 2022 in a depth of 625 m of water approximately 40 km off the shore of Humboldt County (40.97°N, 124.5907°W). A large wave event in December 2020 disabled the Humboldt buoy power supply, resulting in a significant data availability gap during the first year of deployment (Figure 2). Additionally, despite the earlier start dates of the Morro Bay and Humboldt deployments, issues with the buoys' inertial measurement unit were not resolved until 17 October 2020. The final periods of record employed in this study are 17 October 2020 to 30 September 2021 (i.e., almost a year) for Morro Bay and 17 October 2020 to 27 December 2020 and 24 May 2021 to 30 September 2021 (i.e., almost seven months) for Humboldt in order to align with the model data availability discussed in Sect. 2.3. A comprehensive discussion of the California lidar buoy deployments and data availability is provided in Krishnamurthy et al. (2023).

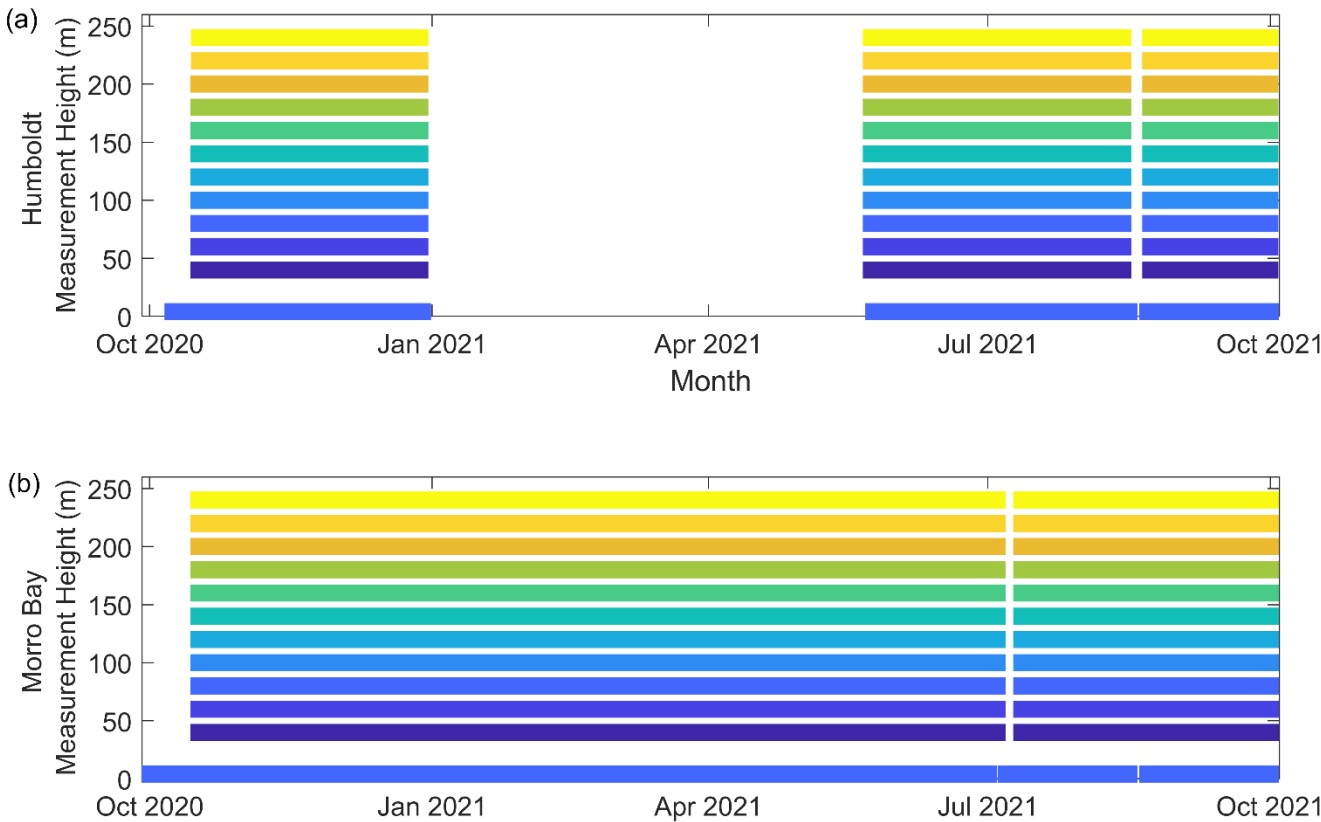

**Figure 2.** Availability of the near surface and lidar wind data at (a) Humboldt and (b) Morro Bay during the study period.

### 2.1 Buoy instrumentation and observations

The DOE lidar buoys were equipped with identical instrumentation for the California deployments. A thorough description of the instrumentation aboard the buoys is available in Severy et al. (2021) and a comprehensive analysis of the data availability, quality, and trends is provided in Krishnamurthy et al. (2023). The most relevant instruments to the LLJ analysis are the Leosphere WindCube 866 lidars for measuring the wind profile and Gill WindSonic ultrasonic anemometers, Vector Instruments A100R cup anemometers, and Vector Instruments W200P wind vanes for measuring the near-surface winds at 4 m a.s.l. Additional near-surface metocean observations that contribute to this analysis include air temperature and relative humidity from a Rotronic MP101A humidity-temperature meteorological probe and air pressure from an RM Young 61302V barometer. The temporal resolutions of the lidar and near-surface measurements utilised in this analysis are 10-minute averages. Quality control of the measurements, discussed in detail in Krishnamurthy et al. (2023) included making sure sensors were not reporting beyond the manufacturer limits, comparisons with nearby sensors, removal of abnormal spikes in the data, physics-based analyses, motion correction, and filtering based on the signal-to-noise ratio for the lidars.

The lidars aboard the buoys retrieved wind measurements at intervals of 20 m between 40 m and 240 m a.s.l. (i.e., 11 height levels). Therefore, the lidar observations employed in this work are valuable for analysing the lowest offshore LLJs but are not comprehensive since LLJ core heights are typically located between 50 m and 300 m a.s.l. (Hallgren et al., 2020).

## 2.2 Observation-based near-surface stability estimates

Atmospheric stability is a significant influence on the wind profile, including the occurrence of LLJs, and therefore impacts the amount of possible energy that can be produced by a wind turbine. To assess the relationships between atmospheric stability near the surface and LLJs higher in the atmosphere, the Obukhov length $L$ is estimated using the Tropical Ocean-Global Atmosphere Coupled Ocean-Atmosphere Response Experiment (COARE) version 3.6 wind speed-based algorithm using the near-surface lidar buoy metocean observations (Fairall et al., 1996; Edson et al., 2013; Sauvage et al., 2023). Typically, $L$ is defined as

$$L = -\frac{\overline{T_v} \cdot u_*^3}{k \cdot g \cdot \overline{w'T_v'}} \tag{1}$$

where $T_v$ is the virtual temperature, $u_*$ is the friction velocity, $k$ is the von Kármán constant, $g$ is gravitational acceleration, and $\overline{w'T_v'}$ is the kinematic virtual temperature flux. To compute the traditional measure of surface layer stability, $z/L$, heat and momentum turbulent fluxes are needed, but measurements of these fluxes are not available to us. The developers of the COARE series of parameterizations provide iterative algorithms that relate these fluxes to measured mean state thermodynamic and wind fields in a self-consistent manner that is also consistent with Monin-Obukhov similarity theory. The COARE parameterizations are specifically adapted to the ocean environment, for which the wave state must either be provided or parameterized as a function of wind speed so that the turbulent momentum flux may be determined. Multiple marine observational datasets of momentum flux have been used by the COARE developers over the years to determine and refine these relationships for general global applications.

## 2.3 Wind resource characterization models

In 2020, NREL produced a 20-year wind resource assessment for the California Pacific Outer Continental Shelf (CA20) using the WRF model in conjunction with the MYNN PBL scheme (Nakanishi and Niino, 2006; Optis et al., 2020). CA20 was validated with coastal radars and near-surface buoy observations. However, when compared with the two DOE floating lidar buoys, large positive wind speed biases across the vertical wind speed profile were noted (Liu et al., 2023). To assess possible sources of error in the CA20 model, extensions to the original 20-year dataset were run using the original CA20 model setup with two PBL schemes: the original MYNN scheme (CA20-Ext) and the YSU scheme (Hong et al., 2010), the latter of which was ultimately incorporated into NREL's 2023 National Offshore Wind data set (NOW-23). Bodini et al. (2022) and Liu et al. (2023) found that the CA20-Ext wind speeds had a large positive bias relative to the DOE lidar buoy observations due to a coastal warm bias in the MYNN simulation during synoptic scale northerly flows and a tendency in the

MYNN simulation to overestimate the occurrence of stable conditions. The YSU-based setup was adopted to produce a revised long-term (2000 – 2022) numerical dataset for offshore California, which is part of NOW-23 (Bodini et al, 2023). CA20-Ext and NOW-23 include 61 vertical layers and, as CA20 extensions to evaluate the PBL schemes, output wind estimates at 10 m and every 20 m between 20 m and 200 m a.s.l. at 5-minute temporal resolution and 2-km horizontal resolution. In this work, CA20-Ext and NOW-23 are evaluated during the 1-year period of 1 October 2020 – 30 September 2021 for which CA20-Ext was run to assess their performance in LLJ representation.

Additionally, we examine the performance of ERA5, a popular global reanalysis product for wind energy resource assessments, in representation of LLJs off the California coast. In addition to being a widely utilized tool for wind energy analysis, ERA5 is also the source of the lateral boundary conditions for CA20 (Optis et al., 2020) and NOW-23 (Bodini et al., 2023). ERA5 provides extensive temporal coverage from 1950 through present time at 1-hour temporal resolution (Hersbach et al., 2020). Between 1 October 2020 and 30 September 2021, ERA5 was found to overestimate the Humboldt lidar buoy observations at 100 m a.s.l. by 0.4 m s$^{-1}$ and underestimate the Morro Bay lidar buoy observations at 100 m a.s.l. by 0.4 m s$^{-1}$ (Sheridan et al., 2022). ERA5 includes 137 vertical layers, of which the lowest ten and nine that cover the surface up to approximately 200 m a.s.l. are examined at Morro Bay and Humboldt, respectively, in the following LLJ study in order to align with the vertical coverage of CA20-Ext. The vertical intervals of ERA5 are smallest near the surface and increase with height, ranging from 22 m to 42 m at Morro Bay and 24 m to 41 m at Humboldt.

**2.4 Low-level jet identification**

An LLJ is a local wind speed maximum that occurs in the lowest heights of the atmosphere, but no universal methodology exists for quantifying LLJs (Hallgren et al., 2020). While the recent research of Hallgren et al. (2023) suggests defining LLJs according to their shear, LLJ definitions most commonly employ a threshold for fall-off, which is the difference between the maximum (the jet core speed) and the subsequent (moving upwards in altitude) local wind speed minimum (Kalverla et al., 2019). Using wind speed profiles over the lowest 1.25 km, Carroll et al. (2019) consider an LLJ fall-off threshold of 5 m s$^{-1}$. Nunalee and Basu (2013) employ a fall-off threshold of 4 m s$^{-1}$ above and below the jet core using wind profiles up to 2 km in height. Using wind profiles within the lowest hundreds of meters, Kalverla et al. (2019) focus the majority of their LLJ analysis using a threshold of 2 m s$^{-1}$, whereas Hallgren et al. (2020) categorize weak and strong LLJs using thresholds of 1 m s$^{-1}$ and 2 m s$^{-1}$, respectively. Aird et al. (2022) identify LLJs if the maximum speed in the wind profile deviates from any speeds above and below by at least 2 m s$^{-1}$ and 20%. Debnath et al. (2021) consider a wind profile to be an LLJ if the wind speed gradient between the bottom of the profile and the jet core exceeds a threshold shear value of 0.035 s$^{-1}$ and the wind speed fall-off above the core is at least 1.5 m s$^{-1}$ and 10% of the core speed. McCabe and Freedman (2023) define an LLJ as having a shear exponent at least ±0.2 above and below the jet maximum, along with a wind speed threshold based on the mean wind speeds in the rotor plane during days with sea breeze events. To provide consistency with similar LLJ studies that utilise wind profiles within the lowest hundreds of meters of the atmosphere, this work employs a fall-off threshold of 2 m s$^{-1}$ above and below the core speed to define an LLJ. No restrictions are placed on the vertical distance between the jet core and

the heights at which the fall-off threshold is achieved within the limits of the observations (4 m to 240 m) and models (10 m to 200 m).

## 3 Offshore low-level jet observations

### 3.1 Characteristics of observed offshore LLJs

Of the 24,878 observations at 10-minute resolution where at least 75% of the lidar wind profile was available during the Humboldt deployment, 781 (3%) had LLJ occurrences. At Morro Bay, 47,906 observations had ≥ 75% lidar wind profile availability and 1,995 (4%) had LLJ occurrences. Figure 3 displays the distributions of LLJ observations according to core height, with 140 m a.s.l. being the most frequently observed LLJ core height across both buoy deployment locations. It is important to recall that the frequencies of LLJ occurrence for the highest heights of this analysis (≥ 200 m) are expected to be underrepresented due to the limitations of the lidar retrieval extent.

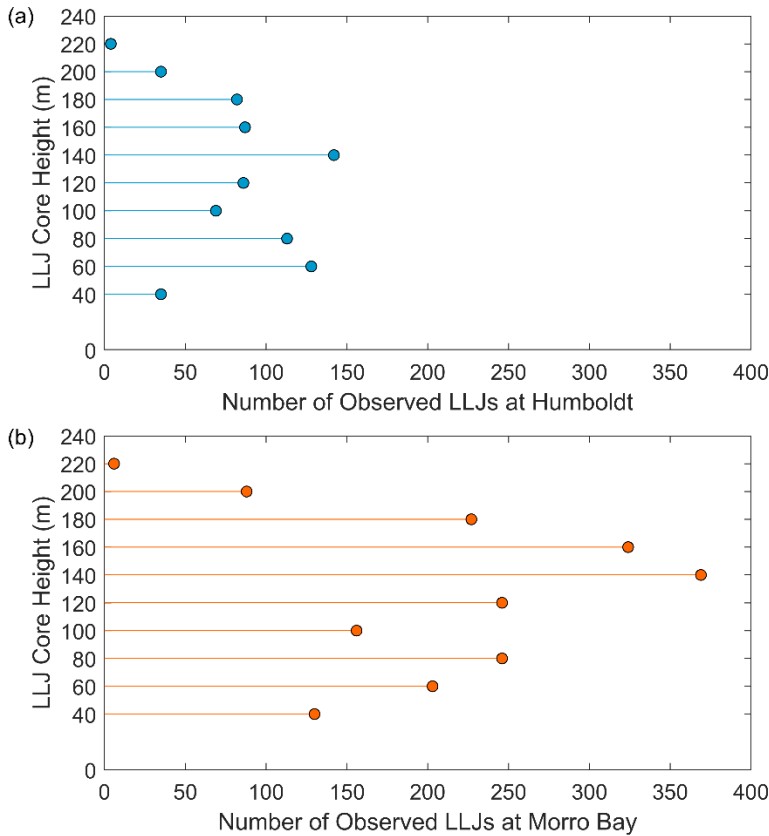

**Figure 3.** Distributions of observed LLJs occurring over 10-minute intervals by height a.s.l. at (a) Humboldt and (b) Morro Bay.

The individual instances of LLJs that occurred in the 10-minute resolution observations can be quantified into LLJ events of varying duration (Figure 4). Here, an LLJ event is defined by consecutive timestamps with LLJs present surrounded by time periods of at least 1 hour when no LLJs are present. At Humboldt, 92 LLJ events occurred for durations ranging from 10 minutes to 10.5 hours. The average (median) LLJ duration at Humboldt was 1.6 hours (0.8 hours). At Morro Bay, 168 LLJ events were documented, with durations ranging from 10 minutes to 22.5 hours. The average (median) LLJ duration at Morro Bay was 2.2 hours (1.0 hour).

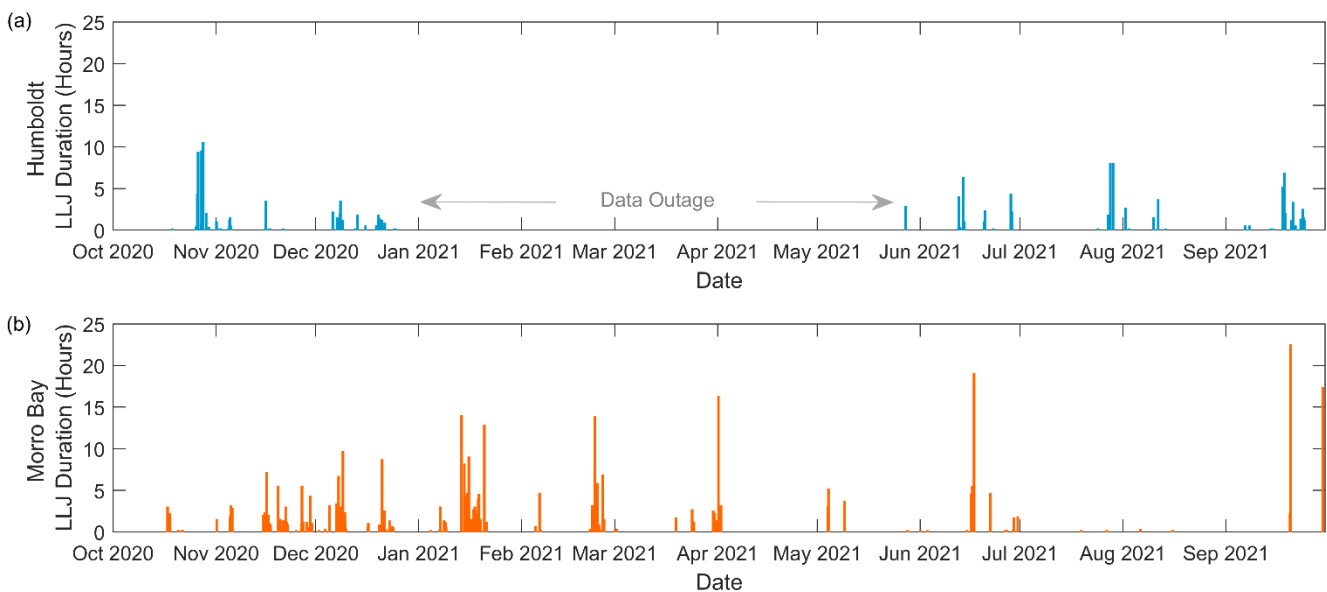

**Figure 4.** Duration in hours of observed LLJs during the (a) Humboldt and (b) Morro Bay lidar buoy deployments.

At both deployment locations, LLJ events occurred in every month with available data coverage (Figure 4). The outage for the Humboldt lidar buoy left two seasons for comparison, with findings of 43 LLJ events (391 10-minute instances) in the fall (September–November) versus 21 LLJ events (256 instances) in the summer (June–August). More LLJ events occurred at Morro Bay during the winter (December–February) (80 events, 976 instances), followed by fall (45 events, 571 instances), spring (March–May) (22 events, 243 instances), and summer (20 events, 205 instances). These findings for offshore LLJs closest to the surface are seasonally in contrast with the higher-altitude California coastal jet (core heights of 300 m – 400 m) that occurs in the summertime due to the presence of the North Pacific High (Parish 2000). It is important to recall that the observational timeseries began on 17 October 2020; therefore, the month of October is potentially underrepresented in LLJ frequency.

At Morro Bay, a dependency on the time of day is noted for LLJ presence, with more LLJs occurring in the hours before local midnight and fewer LLJs occurring during the morning, afternoon, and early evening (Figure 5). At Humboldt, however, LLJs occurred across the diurnal cycle with little variation in frequency (Figure 5), though the comparison may be biased due to the data outage during the Humboldt deployment.

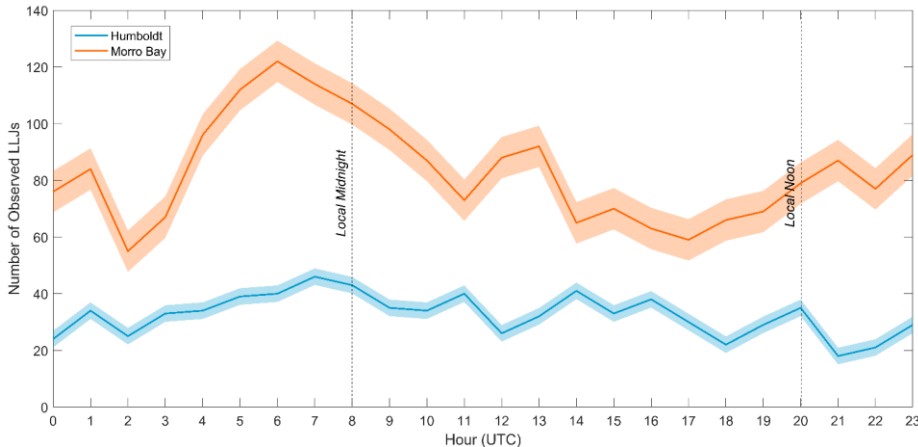

**Figure 5.** Diurnal trends in LLJ occurrences during the Humboldt and Morro Bay lidar buoy deployments with 95% confidence intervals.

The occurrence of offshore LLJs at the lidar buoy deployment locations 40 km off the California coastline are strongly related to the wind flow direction relative to land proximity (Figure 1). At Humboldt, the predominant wind flow is from the north, 350°–10°, with a secondary southerly component, 170°–180° (Figure 6a). During LLJ events, the rotor layer winds at Humboldt tend to be land-based, from the north-northeast (0°–20°) and Cape Mendocino to the south-southeast (160°-180°), though some LLJ events are associated with the offshore flow directions of 180°-220° (Figure 6c). At Morro Bay, the predominant wind flow is from the north-northwest, 330°–340° (Figure 6b). During LLJ events at Morro Bay, the winds are oriented from Point Sur to the north, 340°–0° (Figure 6d).

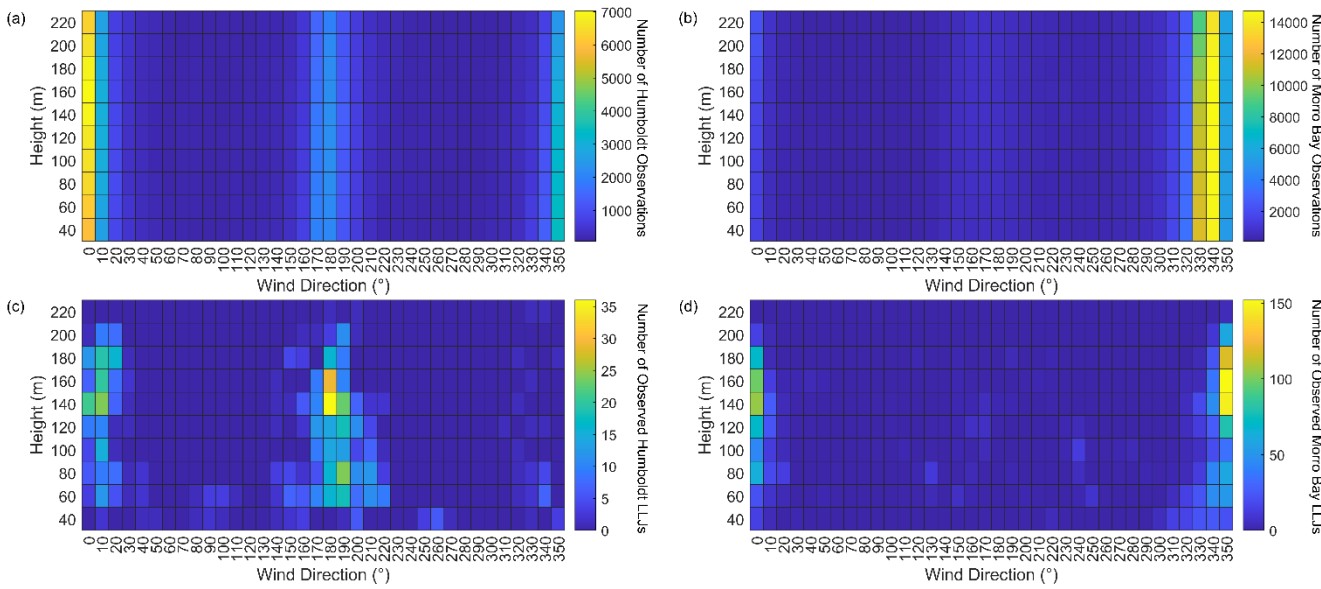

**Figure 6.** Distributions of (a), (b) total observations and (c), (d) observed LLJs by wind direction and height a.s.l. during the Humboldt (left) and Morro Bay (right) buoy deployments between 1 October 2020 and 30 September 2021.

LLJ core wind speeds ranged from 3.8 m s$^{-1}$ to 28.1 m s$^{-1}$ at Humboldt and from 3.3 m s$^{-1}$ to 26.0 m s$^{-1}$ at Morro Bay. Relating the analysis to wind turbine power curves, these wind speeds traverse the cubic portion, the maximum power production portion, and the cut-out or gradual derating portion. Median core speeds at Morro Bay increased steadily with jet height, while Humboldt median core speeds did not deviate as significantly with jet core height (Figure 7). At most measurement heights, the median LLJ core speeds exceed the median deployment-wide wind speeds. The differences between the median LLJ core and deployment-wide wind speeds are especially pronounced at Morro Bay (e.g., differences exceeding 10 m s$^{-1}$ at heights of 200 m and above), where the wind profiles tend to have less shear than those from the Humboldt deployment (Sheridan et al., 2022).

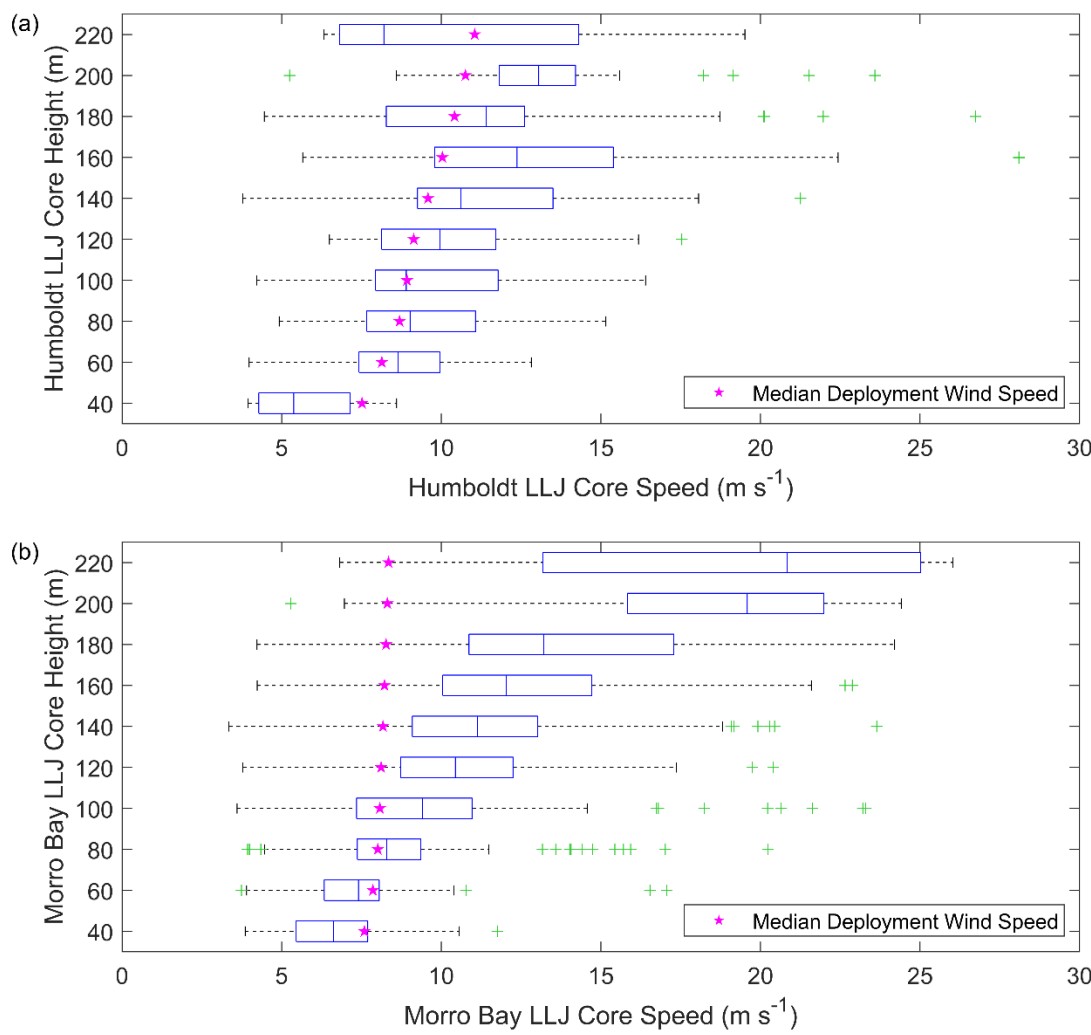

**Figure 7.** Number of observed LLJs by wind speed and height a.s.l. during the (a) Humboldt and (b) Morro Bay buoy deployments, between 1 October 2020 and 30 September 2021. The median error metrics are indicated with the blue line inside each box; the 25th and 75th percentiles form the blue boxed range; the minimum and maximum wind speeds, excluding outliers, form the black whiskers; and

outliers are indicated with green plus signs. Magenta stars indicate the median deployment-wide observed wind speeds according to height a.s.l.

## 3.2 Relationships between LLJs and near-surface metocean observations

While near-surface dynamics have a strong role in the generation and characteristics of LLJs, LLJs similarly influence the state of the atmosphere beneath the jet core through the turbulent transmission of momentum to the surface (Parish, 2000). Yang et al. (2023) explored the influence of nocturnal boundary layer LLJs on land-based near-surface observations and confirmed an inverse relationship between bulk wind shear (between the LLJ core and the surface) and veer in near-surface wind direction ($\Delta WD$) over 10-minute intervals due to the degree of shear-generated turbulence near the surface. In both the Yang et al. (2023) study and this analysis, bulk shear is defined as $(u_{jet} - u_{sfc})/(z_{jet} - z_{sfc})$, where $u_{jet}$ and $u_{sfc}$ are the wind speeds of the LLJ core and the near-surface measurements, respectively, and $z_{jet}$ and $z_{sfc}$ are the heights of the LLJ core and the near-surface measurements (4 m in this analysis), respectively.

At the offshore lidar buoy deployment locations, no such trends in $\Delta WD$ over 10-minute intervals were identified (Figure 8), however this could be attributable to the reduced range of bulk shear values observed at Humboldt and Morro Bay (up to 0.10 s$^{-1}$ and 0.15 s$^{-1}$, respectively) as compared with the bulk shear observations of Yang et al. (2023), which reach 0.4 s$^{-1}$. Instances of minimal temporal change in surface wind direction ($\Delta WD \leq 5°$) were well-distributed for bulk shear values ranging from 0.01 s$^{-1}$ to 0.08 s$^{-1}$ at Humboldt and 0.01 s$^{-1}$ to 0.12 s$^{-1}$ at Morro Bay, and large temporal changes in surface wind direction ($\Delta WD \geq 20°$) were similarly well-distributed across the same bulk shear range at Morro Bay (few large temporal changes in surface wind direction were observed during Humboldt LLJs) (Figure 8). For bulk shear values exceeding 0.08 s$^{-1}$ at Humboldt and 0.12 s$^{-1}$ at Morro Bay, all values of $\Delta WD$ were small (within 10°) but the sample size at this range is too limited to draw conclusions.

The deployment-wide assessments of Sheridan et al. (2022) found predominantly near-neutral atmospheric stability ($z/L \approx 0$) at $z = 4$ m a.s.l. and $L$ as defined in Eq. 1 for both Humboldt and Morro Bay. Outside of the predominant near-neutral conditions, Morro Bay tended toward unstable ($z/L < 0$) while the stability parameter was more evenly distributed among unstable and stable values at Humboldt (Sheridan et al., 2022). During LLJ events, conditions near the surface predominantly ranged from near neutral to stable ($z/L > 0$) at both lidar buoy locations, with a small quantity of instances of unstable conditions in conjunction with low bulk shear (Figure 8). For onshore LLJs, Yang et al. (2023) found that weakly stable conditions were associated with small temporal changes in surface wind direction ($\Delta WD \leq 5°$) with little relationship between the two parameters holding with increasing stability aside from very stable conditions having greater potential to cause larger $\Delta WD$. In the offshore setting, small values of $\Delta WD$ at Humboldt were prevalent regardless of stability regime (Figure 8). At Morro Bay, small and large values of $\Delta WD$ are noted across the range of $z/L$ estimates, but with very stable conditions coinciding with large $\Delta WD$, akin to the findings of Yang et al. (2023).

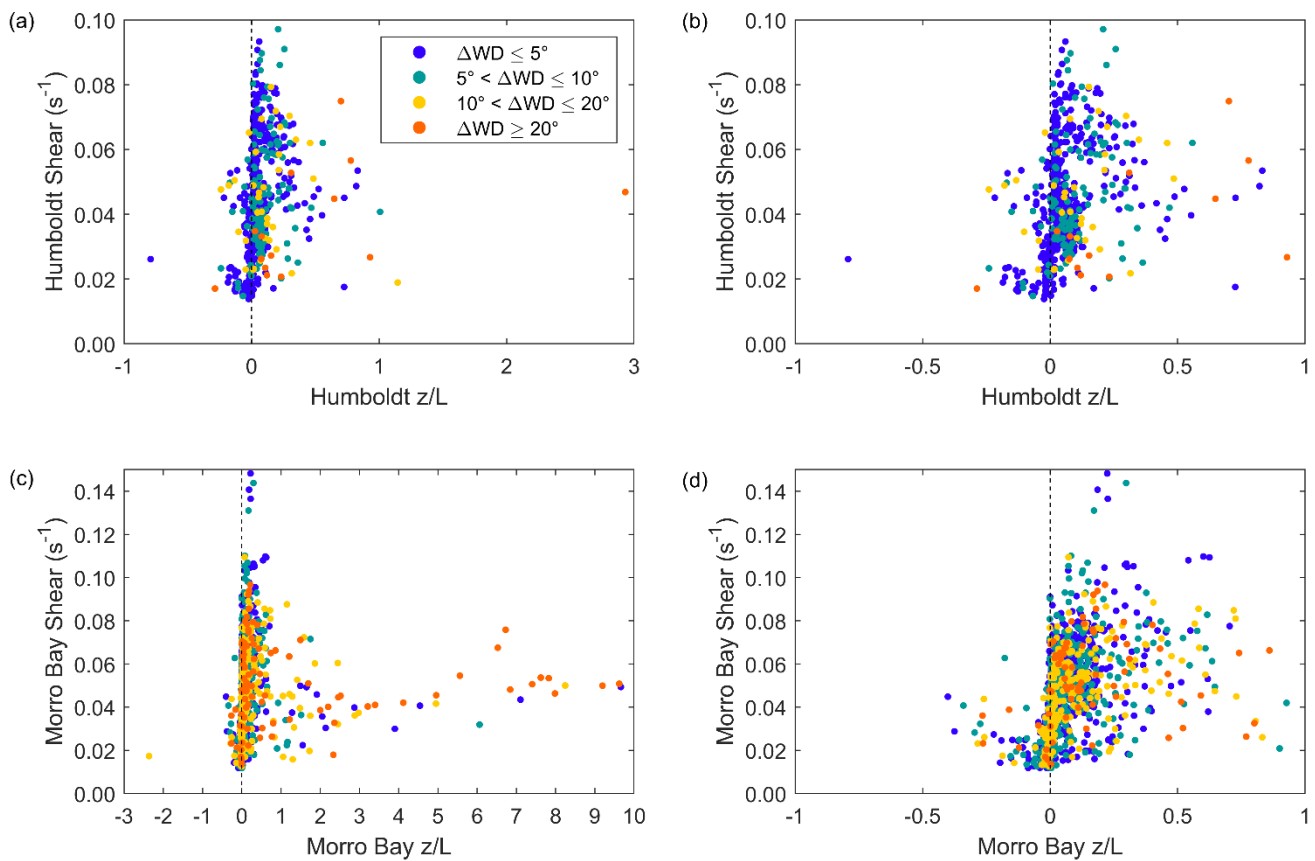

**Figure 8.** Stability parameter, bulk wind shear below the jet core, and 10-minute change in surface wind direction at (a), (b, zoomed in) Humboldt and (c), (d, zoomed in) Morro Bay.

## 4 Model representation of offshore low-level jets

280 In order to compare the performance of LLJ representation in wind models, the lidar buoy observations and model wind data are resampled to include only the top of the hour output to temporally align with the ERA5, which has the coarsest temporal resolution (hourly). As a result, 4,357 timestamps remained for evaluation at Humboldt and 8,081 at Morro Bay. Similarly, we confine the analysis to heights between the surface and the nearest data output height to 200 m, the limiting output height of CA20-Ext. Horizontally, the nearest model grid point to each buoy is selected for evaluation. The Humboldt buoy location

285 is 8.3 km from the nearest ERA5 grid point and 1.1 km from the nearest CA20-Ext and NOW-23 grid points. The Morro Bay buoy location is 10.7 km from the nearest ERA5 grid point and 0.7 km from the nearest CA20-Ext and NOW-23 grid points.

## 4.1 Model accuracy in capturing observed low-level jets

To evaluate the performance of models for LLJ representation off the California coast, we utilize the following methodology from Hallgren et al. (2020) and Kalverla et al. (2020) to categorize whether each model captures, misses, or incorrectly reports an LLJ at a given timestamp. Figure 9 displays the quantities of LLJ hits (timestamps when both the observations and the model indicate a fall-off of $\geq 2$ m s$^{-1}$), misses (timestamps when the observations produce a fall-off of $\geq 2$ m s$^{-1}$ but the model produces a fall-off $< 2$ m s$^{-1}$), false alarms (timestamps when the observational fall-off is $< 2$ m s$^{-1}$ but the model fall-off is $\geq 2$ m s$^{-1}$), and correct rejections (timestamps when both the observational and model fall-offs are $< 2$ m s$^{-1}$). At both buoy deployment locations, CA20-Ext captured the most observed LLJs with 34 hits out of a possible 73 observed LLJs at Humboldt (a 47% success rate) and 102 hits out of a possible 211 observed LLJs at Morro Bay (a 48% success rate). NOW-23 was the next most successful model at LLJ representation with 18 and 28 hits at Humboldt and Morro Bay (25% and 13% success rates), respectively. For comparison, when considering these models at 10-minute resolution, CA20-Ext had success rates of 46% and 50% and NOW-23 had success rates of 26% and 13% at Humboldt and Morro Bay, respectively.

ERA5 did not capture any of the observed LLJs during the observational period. One potential reason for ERA5's lack of success in modelling California offshore LLJs is the challenges the model was found to experience in representing flow reversals at the lidar deployment locations (Sheridan et al., 2022).

While CA20-Ext produced the most LLJ hits for the California deployment, it also produced the most false alarms across the models (150 and 193 at Humboldt and Morro Bay, respectively). The false alarms contribute significantly to the CA23-Ext frequency bias, the ratio of the total number of predicted LLJs to the total number of observed LLJs (Hallgren et al., 2020). A frequency bias of one is a perfect score, while a frequency bias above (below) one indicates model overestimation (underestimation) of the number of observed LLJs. The CA20-Ext frequency biases of 2.5 (Humboldt) and 1.4 (Morro Bay) represent significant model overestimation of observed LLJs, particularly at Humboldt. NOW-23 produced 41 and 70 false alarms at Humboldt and Morro Bay which, combined with the hits, results in frequency biases indicative of model underestimation (0.8 and 0.5, respectively). ERA5 produced 2 and 6 false alarms at Humboldt and Morro Bay, generating frequency biases emblematic of significant model underestimation (<0.1 at both sites).

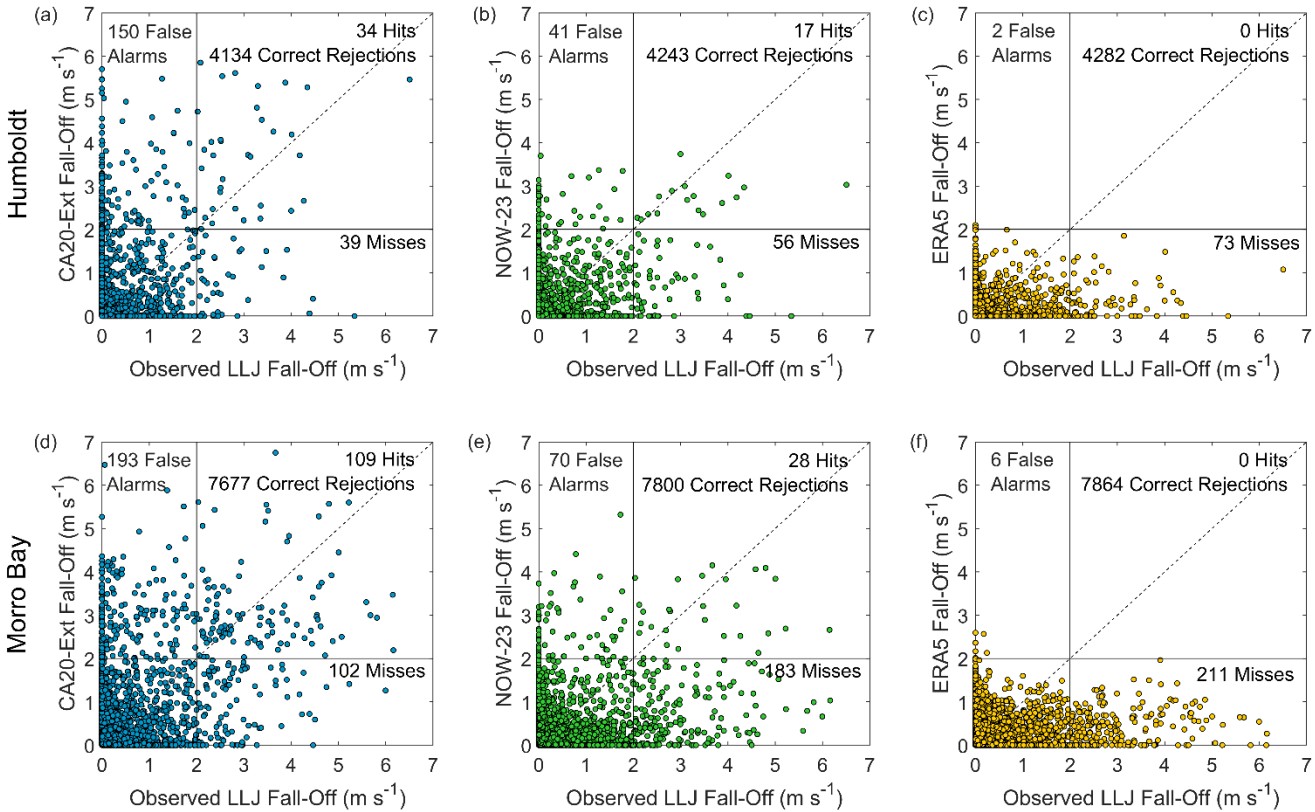

**Figure 9.** Observed versus (a), (d) CA20-Ext, (b), (e) NOW-23, and (c), (f) ERA5 LLJ fall-off at Humboldt (top row) and Morro Bay (bottom row).

For both CA20-Ext and NOW-23, the estimated number of LLJ events is underpredicted relative to the observations, but part of this discrepancy is likely attributable to the increased average duration of LLJ events in both models relative to the observations (Figure 10). As in the earlier observational assessment, an LLJ event is defined by consecutive timestamps with LLJs present at intervals less than or equal to 1 hour. At Humboldt at 10-minute resolution, 62 LLJ events with core heights between the surface and 200 m a.s.l. were documented, as compared with 52 and 27 events simulated by CA20-Ext and NOW-23. While the Humboldt observations classified more LLJ events, both models estimated longer-duration LLJ events than were observed (Figure 10a). The average LLJ event durations for the Humboldt observations, CA20-Ext, and NOW-23, respectively, were 1.4 hours, 3.6 hours, and 2.3 hours. Of the observed Humboldt LLJ events, 63% were of durations less than 1 hour, while 23% of CA20-Ext and 37% of NOW-23 LLJ events were of durations less than 1 hour. No Humboldt LLJ events persisted for 10 hours or more, but 2% of CA20-Ext and 4% of NOW-23 LLJ events persisted for 10 hours or more.

Similar trends occurred for Morro Bay, where 128 LLJ events with core heights between the surface and 200 m a.s.l. were documented, as compared with 88 and 42 events simulated by CA20-Ext and NOW-23. Again, while the Morro Bay observations classified more LLJ events, both models estimated longer-duration LLJ events than were observed (Figure 10b). The average LLJ event durations for the Morro Bay observations, CA20-Ext, and NOW-23 were 1.9 hours, 3.5 hours,

and 2.3 hours. Of the observed Morro Bay LLJ events, 50% were of durations less than 1 hour, while 25% and 36% of CA20-Ext and NOW-23 LLJ events, respectively, were of durations less than 1 hour. At the other extreme, 3%, 9%, and 2% of Morro Bay observed, CA20-Ext-simulated, and NOW-23-simulated LLJ events were documented as persisting for 10 hours or more.

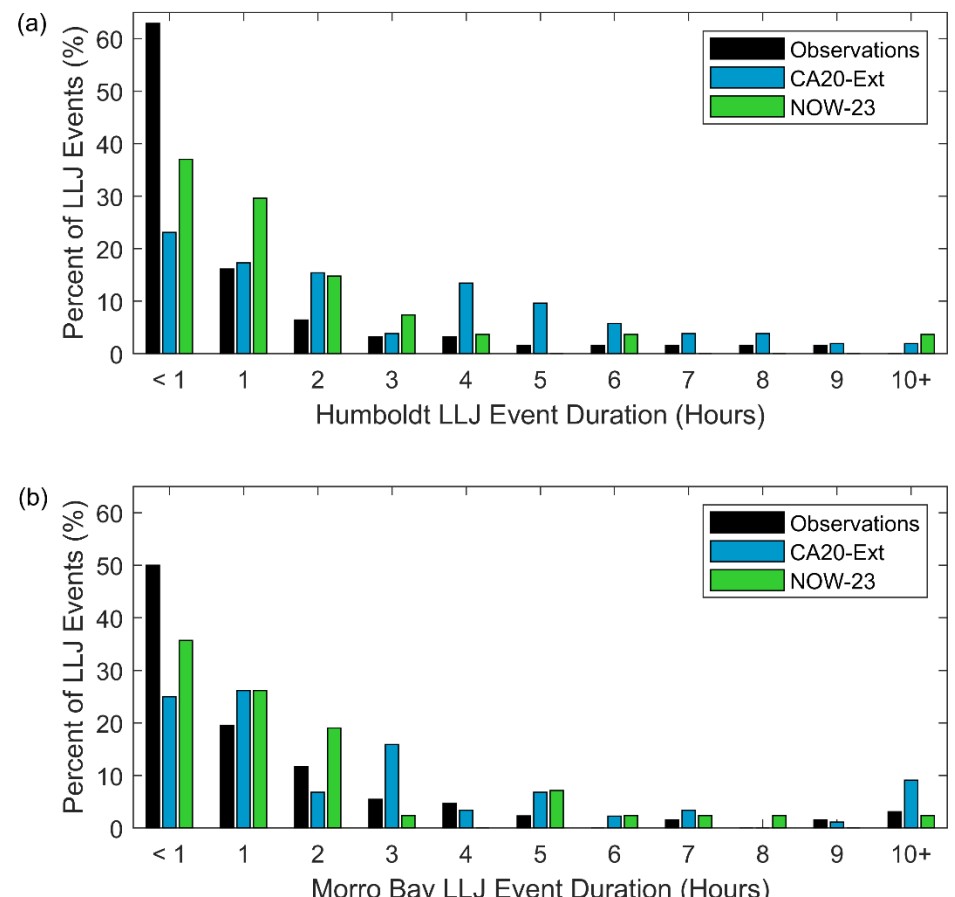

**Figure 10.** Duration of observed and simulated LLJ events at (a) Humboldt and (b) Morro Bay.

**4.2 Timing accuracy of simulated low-level jets**

Common contributors to model wind speed error are timing offsets between observed and predicted atmospheric phenomena (Bianco et al., 2016). Such errors in a forecast framework can significantly impact the economics and reliability of energy trading and grid balancing. While the three models evaluated in this work have challenges in accurately representing the occurrence of LLJs as discussed in the prior section, a prevalence of model timing accuracy is noted for CA20-Ext and NOW-23 when the models do produce LLJs in the temporal vicinity of observed LLJs (Figure 11). At Humboldt, 74% of CA20-Ext LLJs and 53% of NOW-23 LLJs that were simulated within ±3 hours of an observed LLJ occurred at zero hours offset from the observed LLJs. Similarly at Morro Bay, 66% of CA20-Ext LLJs and 46% of NOW-23 LLJs that were

simulated within ±3 hours of an observed LLJ occurred at zero hours offset from the observed LLJs. ERA5 produced no LLJ hits, but it is interesting to note that half of the LLJ false alarms (one of two at Humboldt and three of six at Morro Bay) occurred within ±3 hours of an observed LLJ.

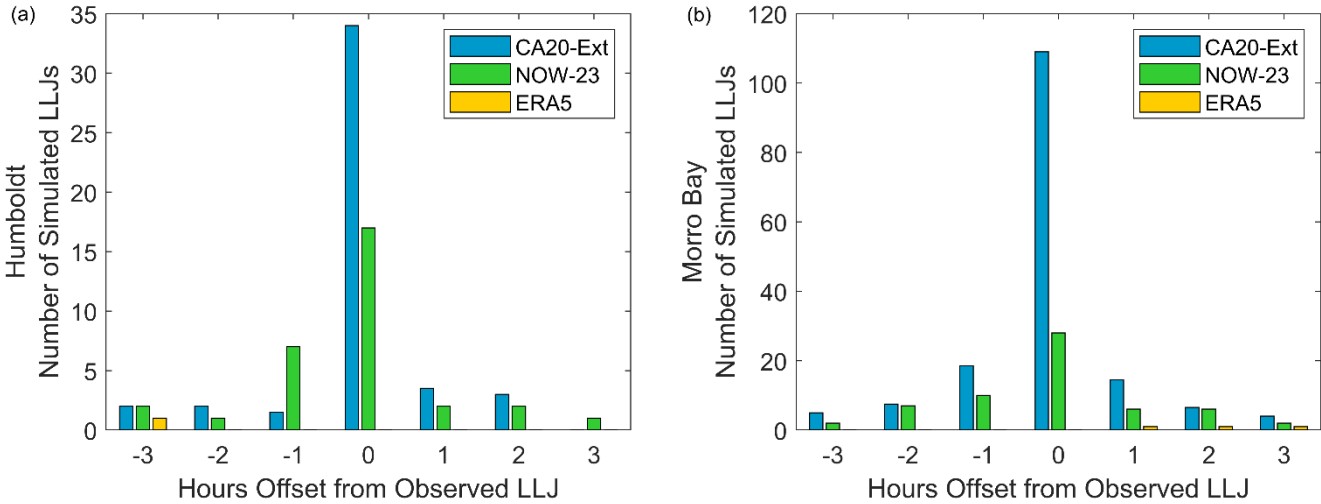

**Figure 11.** Number of hours between simulated and observed LLJs at (a) Humboldt and (b) Morro Bay.

### 4.3 Seasonal accuracy of simulated low-level jets

Since ERA5 was unsuccessful at capturing observed LLJs off the California coast, we remove it in advance of the subsequent analyses. The removal of ERA5 allows us to increase the temporal resolution (and therefore sample size) of the observations and remaining models, CA20-Ext and NOW-23, to 10 minutes.

Despite determining that models underrepresented the occurrence of LLJs in the North Sea, Kalverla et al. (2019) found that their analysis suite of regional and global reanalyses agreed with their observations in terms of seasonal LLJ patterns. Performing a similar analysis using CA20-Ext and NOW-23 at Humboldt and Morro Bay produced inconsistent levels of monthly accuracy according to model and location. Figure 12 shows the percentages of observed and simulated LLJs by month at each deployment location. At Humboldt, CA20-Ext and NOW-23 perform similarly for the seven months of available observations with correlations of 0.8 as compared with the monthly observed LLJ frequencies. Both models and the lidar observations agree that October 2020 had the most LLJ instances, with 33%, 36%, and 44% of LLJ instances from the observations, CA20-Ext, and NOW-23, respectively, occurring in this month (Figure 12a,b). Observationally, August 2021 saw the fewest LLJs at Humboldt (1% of the total observed LLJs) within the lowest 200 m, whereas both models selected June 2021 as the month with the fewest LLJs (< 1% of the total simulated LLJs).

In contrast to the similar performance between the models at Humboldt, CA20-Ext and NOW-23 perform quite differently in terms of seasonal LLJ representation at Morro Bay. For the 12 months of available observations, CA20-Ext

produced a correlation of 0.8, showing similar seasonal performance to the counterpart analysis at Humboldt (Figure 12c).

Conversely, NOW-23 significantly disagreed with the observed seasonal trend in LLJ occurrence at Morro Bay, with a correlation of 0.4 (Figure 12d). The observations and CA20-Ext agreed that January 2021 had the most LLJ instances, with 26% and 22% rates of LLJ occurrence, respectively. NOW-23 estimated that June 2021, February 2021, and March 2021 were the most active months for LLJs, with rates of occurrence of 24%, 21%, and 18%, respectively. The months with the lowest rates of LLJ occurrence (< 1%) at Morro Bay were May 2021 (0%), July 2021 (0.1%), and August 2021 (0.3%).

CA20-Ext predicted no LLJs occurring in July 2021 at Morro Bay, while NOW-23 predicted no LLJs occurring in October 2020, April 2021, July 2021, and August 2021.

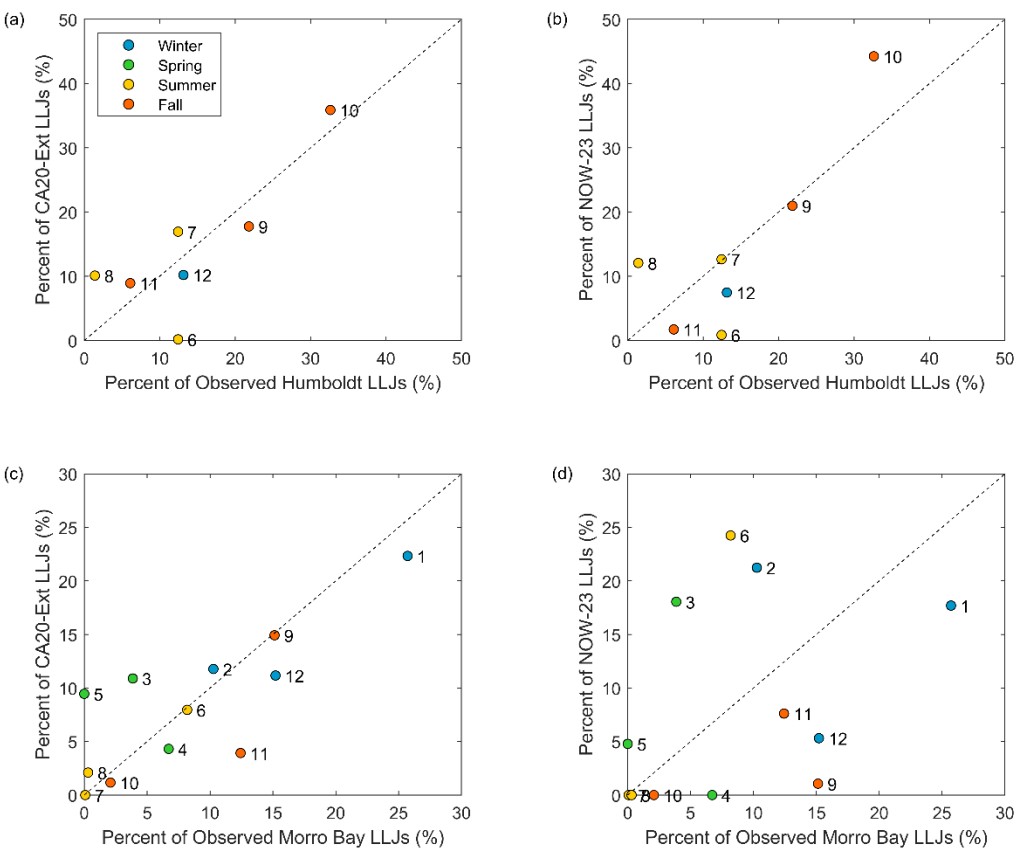

**Figure 12.** Percentages of LLJ instances at 10-minute resolution observed and simulated by (a), (c) CA20-Ext and (b), (d) NOW-23 according to month (numbered) at (a), (b) Humboldt and (c), (d) Morro Bay. The markers are coloured according to season and are 375 labelled by month (1 = January, 2 = February, … 12 = December).

### 4.4 Model bias in low-level jet core height representation

Gadde and Stevens (2021) described the sensitivity of the vertical placement of the LLJ for wind turbine wake recovery, which urges the consideration of bias in simulated LLJ core heights for appropriate wake loss characterization in wind

energy estimates. As presented in Figure 13, both CA20-Ext and NOW-23 underestimate observed LLJ core heights at both buoy deployment locations. At Humboldt, both models present an LLJ height bias of -11 m, however this value is within the 20 m resolution of the observed and modelled datasets. The degree of model LLJ height underestimation is more pronounced at Morro Bay, with CA20-Ext and NOW-23 producing biases of -22 m and -31 m, respectively.

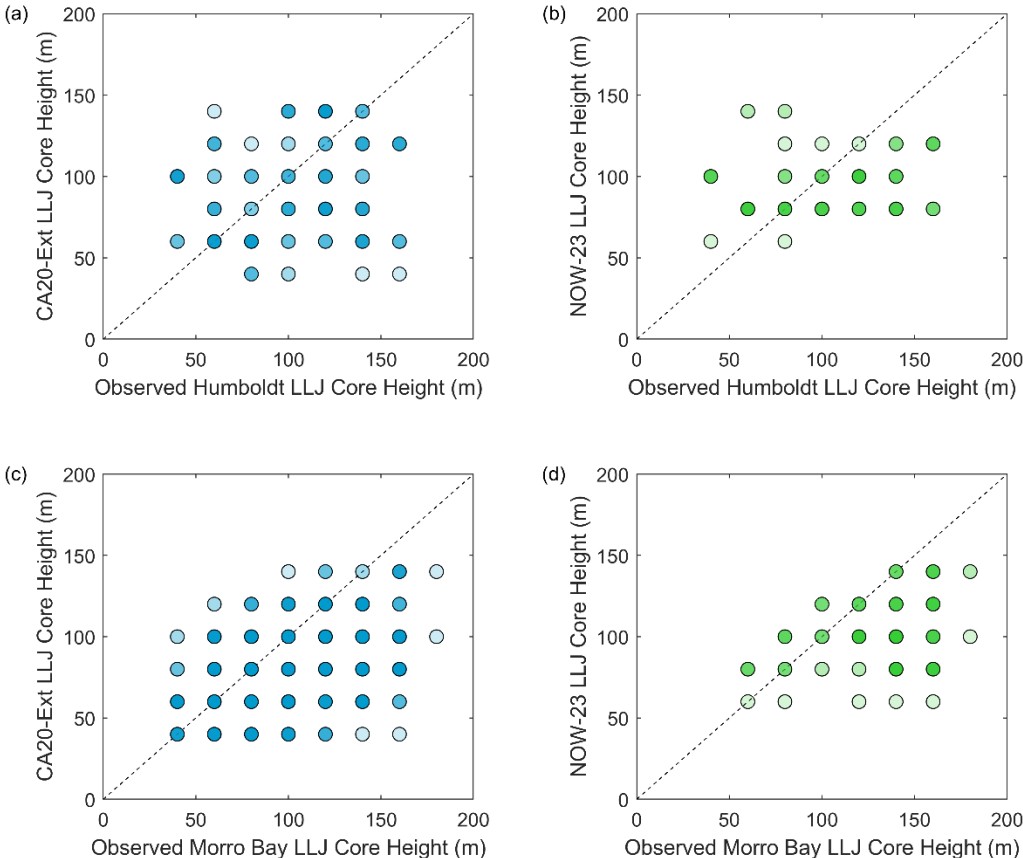

**Figure 13.** Observed versus (a), (c) CA20-Ext and (b), (d) NOW-23 simulated LLJ core heights at (a), (b) Humboldt and (c), (d) Morro Bay. Darker shades indicate higher density of results.

### 4.5 Model bias in low-level jet core speed representation

While CA20-Ext and NOW-23 exhibit negligible LLJ core height bias at Humboldt and negative LLJ core height bias at Morro Bay, model performance differences by location are greater for LLJ core wind speed bias (Figure 14). At Humboldt,

the models tend to overestimate the observed LLJ core speeds, by 1.9 m s$^{-1}$ on average for CA20-Ext and 0.6 m s$^{-1}$ on

average for NOW-23. The CA20-Ext bias at Humboldt during LLJ events is identical to the deployment-wide bias determined by Liu et al. (2023), while the deployment-wide bias for NOW-23 is smaller than for LLJs at 0.1 m s$^{-1}$. At Morro Bay, the models tend to underestimate the LLJ core wind speeds, with biases of -0.6 m s$^{-1}$ and -1.1 m s$^{-1}$ for CA20-Ext and NOW-23, respectively. At both sites and for both models, the largest overestimations of the LLJ core wind speeds occur at the lowest observed core heights, with a trend of biases approaching or achieving underestimation with increasing observed core height.

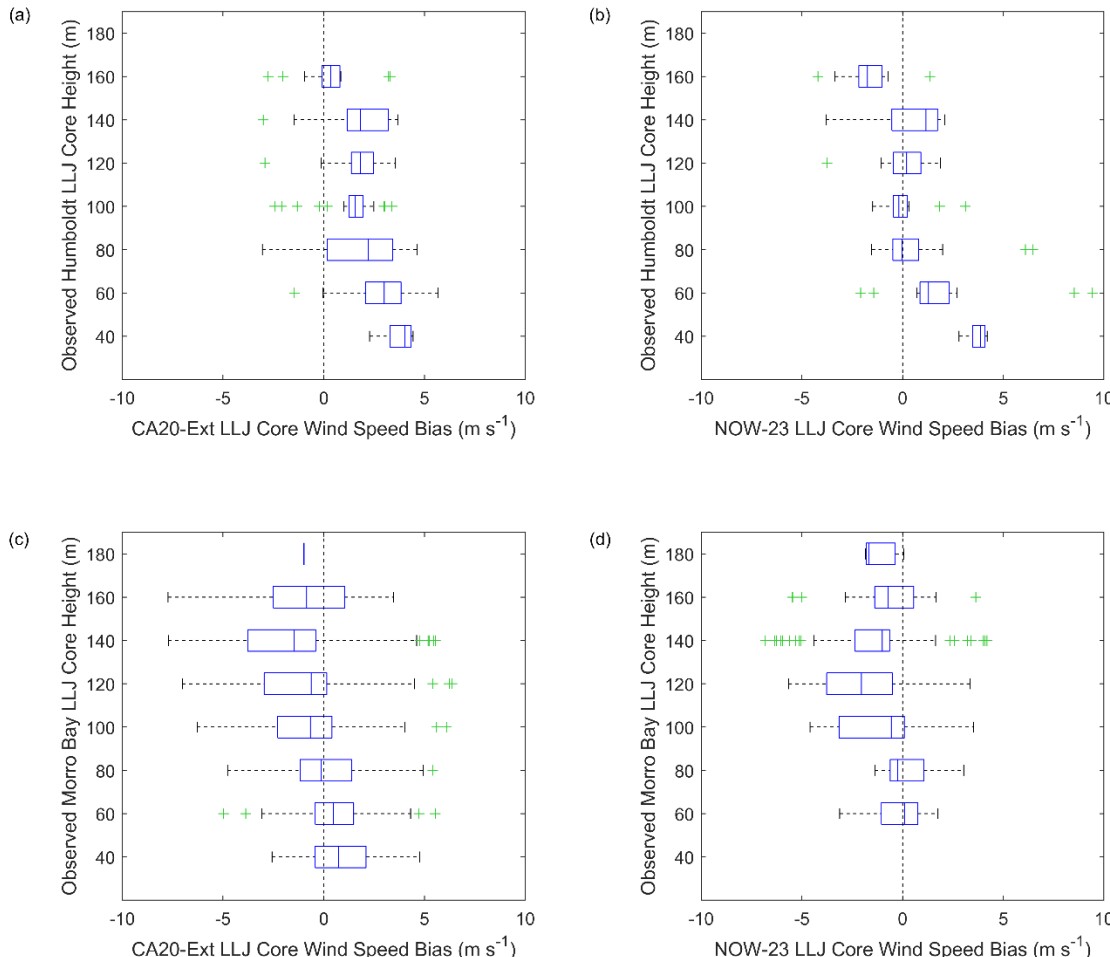

**Figure 14.** Jet core wind speed bias (model wind speed minus observed wind speed) according to observed jet core height for (a), (c) CA20-Ext and (b), (d) NOW-23 simulated LLJ core speeds at (a), (b) Humboldt and (c), (d) Morro Bay. The median error metrics are indicated with the blue line inside each box; the 25th and 75th percentiles form the blue boxed range; the minimum and maximum wind speeds, excluding outliers, form the black whiskers; and outliers are indicated with green plus signs.

Figure 15 provides a case study of an LLJ at Morro Bay that occurred on 15 January 2021 that exemplifies many of the findings of Sect. 4, namely extended model durations of LLJ events and underestimated core heights, along with underestimated core speeds at Morro Bay. The observed LLJ began at 10:00 UTC and ended at 14:10 UTC, a duration of 4.2

hours, and occurs 1.8 hours after a prior LLJ event that began at 7:10 UTC and ended at 8:10 UTC. The CA20-Ext LLJ continues as an extension of the earlier LLJ, which began at 7:30 UTC with no distinct separation between events, as occurs in the observations, and persists through 14:40 UTC. During the observed LLJ duration (10:00 – 14:10 UTC), the maximum observed core speed was 10.8 m s$^{-1}$ while the maximum CA20-Ext core speed was 10.4 m s$^{-1}$. Observed core heights ranged between 60 m and 120 m a.s.l., with an average of 88 m. During the same timeframe, CA20-Ext core heights ranged between 40 m and 60 m a.s.l., with an average of 55 m. While NOW-23 produced periods of faster wind speeds near the surface on 15 January 2021, no timestamps met the LLJ criteria of a fall-off threshold of 2 m s$^{-1}$ above and below the core speed employed in this work.

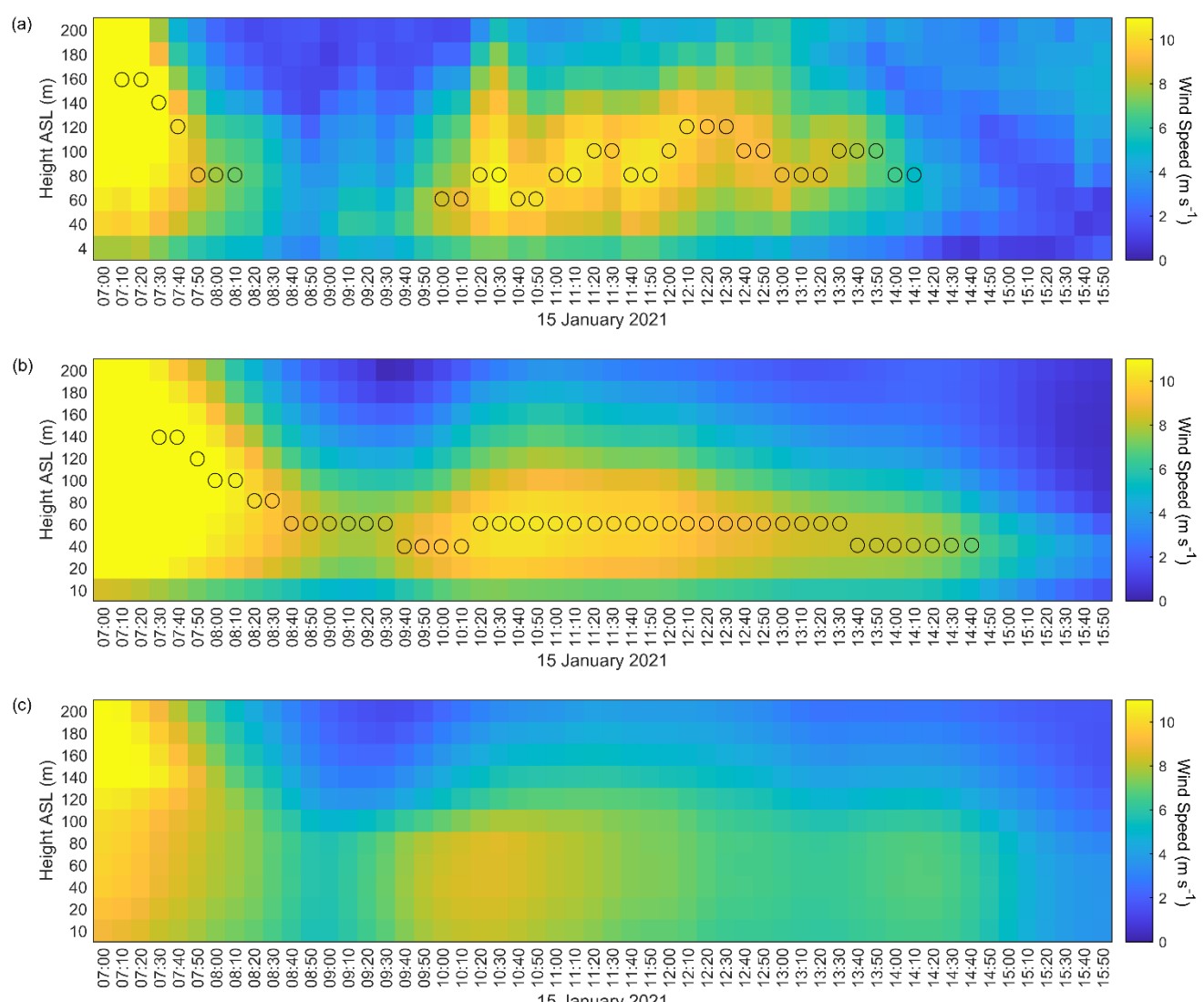

**Figure 15.** (a) Observed, (b) CA20-Ext-simulated, and (c) NOW-23-simulated wind speeds during an observed low-level jet on 15 January 2021 at Morro Bay. Reported timestamps are in UTC. Markers indicate the (a) observed and (b), (c) modelled jet core heights.

## 5 Discussion

Offshore LLJs are complex atmospheric phenomena that impact the wind profile in the marine boundary layer and therefore the amount of potential energy that a wind farm can generate. However, observations of LLJs in offshore settings are temporally and geographically sparse. Two year-long deployments of lidar-mounted buoys in California offshore wind energy areas, Humboldt and Morro Bay, addressed the observational gap for the lowest-occurring LLJs (core heights below 240 m, the extent of the lidar data retrieval). In addition to capturing LLJs, the lidar buoys provided concurrent metocean observations to assess the relationships between LLJs and atmospheric conditions near the surface.

LLJs occurred during 3% and 4% of the Humboldt and Morro Bay study periods, respectively. Despite these small percentages, the impact of LLJs on potential offshore wind farm production could be substantial. As a simple exercise, we simulate wind energy production for a single turbine using the NREL 15 MW offshore wind reference power curve (NREL, 2023) and the rotor equivalent wind speed following the form of Wagner et al. (2014) for two scenarios. First, we apply the power curve to the rotor equivalent wind speeds (calculated using all available observations between the heights $z_i$ of 40 m and 240 m) during instances when LLJs occurred at any height. Second, during the same temporal period as the first scenario, we create rotor equivalent wind speed estimates using power law-based wind speed estimates $u_i$ at the same heights $z_i$ between 40 m and 240 m (Eq. 2). To simulate what the power output would be with no LLJs present, the shear exponents $\alpha$ (Eq. 3) are determined using the observed wind speeds $u$ at the lowest (*lo*) and highest (*hi*) available heights $z$:

$$u_i = u_{hi} \left(\frac{z_i}{z_{hi}}\right)^{\alpha} \tag{2}$$

$$\alpha = \frac{\ln(u_{hi}/u_{lo})}{\ln(z_{hi}/z_{lo})} \tag{3}$$

The NREL 15 MW offshore wind reference power curve has a rotor diameter of 240 m, and we employ a 140 m hub height so that all lidar buoy observed LLJs occur within the rotor-swept plane.

Except for a small amount of cases, the LLJs observed with the lidar buoys yielded higher rotor equivalent wind speeds and simulated wind power than would have occurred without LLJ presence using a power law-based wind profile developed from the observed wind speeds at the lowest and highest available heights (Figure 16). Through the duration of Humboldt deployment, a single 15 MW wind turbine would have produced 0.4 GWh more energy from the observed LLJs than if the LLJs had not occurred. At Morro Bay, a single 15 MW wind turbine would have produced 1.1 GWh more energy from the observed wind profiles than if the LLJs had not occurred.

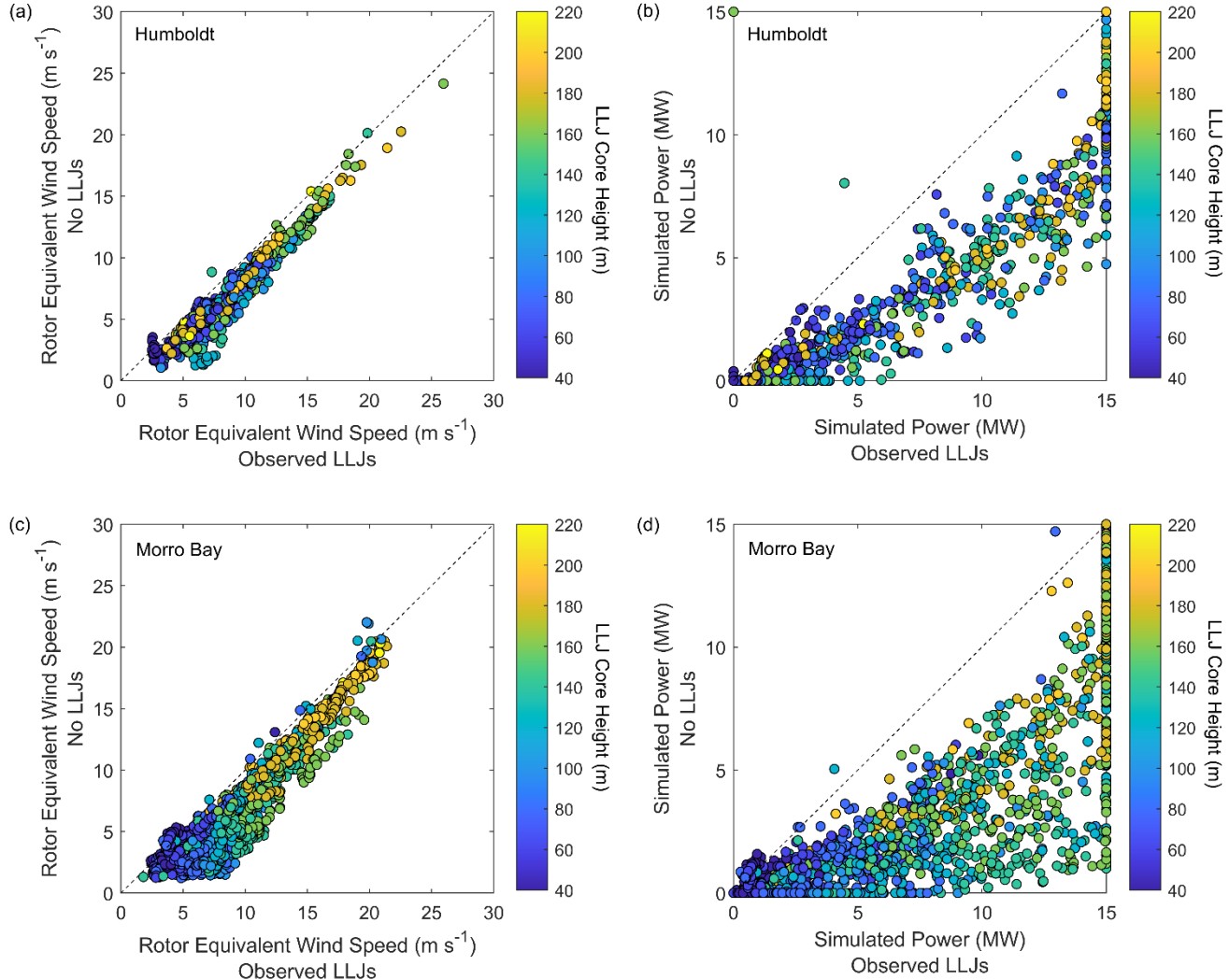

**Figure 16.** Rotor equivalent wind speed during observed LLJs versus simulated power law-based rotor equivalent wind speeds (no LLJs) at (a) Humboldt and (c) Morro Bay. Simulated power using the NREL 15 MW offshore wind reference power curve (NREL, 2023) at (b) Humboldt and (d) Morro Bay using the wind speeds in (a) and (c), respectively.

Representation of LLJs in atmospheric models is important for accurate wind energy generation estimates and planning for short-duration impacts on the electric grid, such as from ramp events. ERA5, a commonly employed reanalysis model for wind energy studies, did not simulate any of the observed LLJ events at Humboldt or Morro Bay, likely due in part to coarse spatial resolution and mishandling of flow reversals. The higher resolution CA20-Ext and NOW-23 datasets yielded marginally better representation of LLJs, with approximately 50% success rates at both deployment locations for CA20-Ext using the MYNN planetary boundary layer scheme versus 25% or less success rates for NOW-23 using the YSU scheme (Figure 9). During instances when CA20-Ext and NOW-23 accurately predicted an observed LLJ occurrence, the simulated

jet core heights tended to be underestimated at Morro Bay. Combining our LLJ case study in Figure 15 with the NREL 15 MW offshore wind reference power curve, we find that despite the similarity of the mean observed and CA20-Ext jet core wind speeds (9.1 m s$^{-1}$ and 9.3 m s$^{-1}$, respectively) during the observed 4.2-hour LLJ event, the shape and vertical placement of the CA20-Ext LLJ would result in a single turbine energy estimate 5.7 MWh less than the observations would indicate, based on rotor equivalent wind speed calculations.

Coastal and offshore measurement campaigns, while challenging to execute, provide valuable data collections to support the evaluation of potential wind energy generation in an offshore setting. The increasing number of such deployments is advantageous for understanding the characteristics of meteorological influences, such as LLJs, on the wind profile in unique locations. For example, recent measurement campaigns yielded locationally-driven diversity in the time of year for most frequent LLJ occurrence, namely May in the Baltic Sea (Hallgren et al., 2020), April – November in the New York Bight (McCabe and Freedman, 2023), and January at Morro Bay. Offshore observations are also needed for highlighting research areas for wind modelling improvement, such as the studies of Hallgren et al. (2020) and this work in noting the limitations of ERA5 representation of LLJs in distinct environments. The breadth of wind profile characteristics revealed by such measurement campaigns encourages similar analyses in new areas of offshore wind development interest. Subsequent DOE lidar buoy deployments include the waters off Hawaii and the U.S. Atlantic coast. Additionally, we look forward to expansion in the understanding of offshore LLJ occurrence and features, particularly in vertical extent, as floating lidar technology continues to advance. We hope this work encourages increased offshore wind observational campaigns to support validation and improvements for modelling of atmospheric phenomena like LLJs.

**Code and data availability**

The lidar buoy utilized in this study are freely and publicly available from the U.S. Department of Energy. The Humboldt lidar and near-surface observations are available at https://doi.org/10.21947/1783809 (U.S. Department of Energy, 2023a) and https://doi.org/10.21947/1783807 (U.S. Department of Energy, 2023c), respectively. The Morro Bay lidar and near-surface observations are available at https://doi.org/10.21947/1959721 (U.S. Department of Energy, 2023b) and https://doi.org/10.21947/1959715 (U.S. Department of Energy, 2023d), respectively. ERA5 is available through the Copernicus Climate Change Service Climate Data Store at http://cds.climate.copernicus.eu (Copernicus, 2022). The CA20 extensions were developed internally to assess the impact of PBL scheme selection on the performance of the original version of CA20 (Optis et al., 2020) and supported the development of the publicly-available Pacific Ocean NOW-23 datasets (Bodini, 2023).

**Author contributions**

Research conceptualization, manuscript preparation, software development, and analysis were performed by L. Sheridan. Buoy deployment management, research conceptualization and team management, and software development were performed by R. Krishnamurthy. Data were obtained and processed by W. Gustafson Jr., Y. Liu, B. Gaudet, N. Bodini, R. Newsom, and M. Pekour. All authors contributed to the manuscript edits and technical review.

**Acknowledgements**

This work was authored by the Pacific Northwest National Laboratory, operated for the U.S. Department of Energy by Battelle (contract no. DE-AC05-76RL01830). This work was authored in part by the National Renewable Energy Laboratory, operated by Alliance for Sustainable Energy, LLC, for the US Department of Energy (DOE) under contract no. DE-AC36-08GO28308. Funding was provided by the US Department of Energy Office of Energy Efficiency and Renewable Energy Wind Energy Technologies Office. The views expressed in the article do not necessarily represent the views of the DOE or the US Government. The US Government retains and the publisher, by accepting the article for publication, acknowledges that the US Government retains a nonexclusive, paid-up, irrevocable, worldwide license to publish or reproduce the published form of this work, or allow others to do so, for US Government purposes. The authors would like to thank Shannon Davis and Mike Derby at the U.S. Department of Energy Wind Energy Technologies Office for funding this research and Doug Boren, Necy Sumait, and Frank Pendleton at the U.S. Department of the Interior Bureau of Ocean Energy Management for funding the Humboldt and Morro Bay deployments. The authors would also like to thank Alyssa Matthews and the three anonymous reviewers for their thoughtful suggestions for manuscript improvement.

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
