# Peer review of "Offshore low-level jet observations and model representation using lidar buoy data off the California coast"

_Wind Energy Science, 2023_

## Referee Comment (RC2)

**Summary Comments:**

This study presents a nice comparison of observations from a temporary LiDAR buoy deployment off of the California coast, to the NREL 20-year Wind Resource Dataset (CA20-Ext), the 2023 National Offshore Wind Dataset (NOW-23), and the ERA5 reanalysis product. This work is important as we look forward to offshore wind development on the West Coast, as model errors in low-level jet (LLJ) representation can affect wind turbine energy generation. This study successful compares three different models to offshore lidar observations and identifies the strengths and weaknesses of each.

I do have a few minor comments that I believe will help the clarity and flow of the paper. Some general comments are that I do think that the figures could benefit from panel titles that have the lidar buoy location/model name. Even though it is explained in the figure caption, I think it would be nice to also add titles for quick referencing. I also noticed that the word "respectively" was very used often in the analysis sections (specifically sections 4.1 and 4.2), as well as a few long run-on sentences which made the paper more difficult to read. The rest of my comments are more specific and are listed below.

Overall, I think this is an excellent study and I am excited to see the results of another offshore lidar deployment. It is difficult to get profiler measurements offshore, and the field needs studies like this one, that can showcase the discrepancies between model predictions and observational data for wind energy applications. I really appreciated how the authors compared how the model bias in LLJ prediction could impact wind energy generation forecasts. I recommend this paper goes to publication.

**Major Comments:**

Lines 30-35: This paper is missing a figure that shows entire study region, including the location of the 2 lidar buoys (at Morro Bay and Humboldt) as well as the proposed 5 lease areas. I think this would be very helpful in providing some context for those that aren't familiar with the study area.

Lines 85-90, 188: Again, referencing a map of the study region here would help your discussion.

Lines 153-156: Why did you choose a 2 ms$^{-1}$ windspeed drop off threshold in your LLJ identification? Is this based off a percentage of the mean wind speed that is typically seen in the region? I think explaining why the 2 ms$^{-1}$ works well in this offshore environment will be helpful in making this study more applicable in other areas.

Lines 159-165: Did you consider any other variables when identifying the LLJs? You mention that there is no restriction set on the vertical distance of the 2ms$^{-1}$ fall off, but did you set a minimum or maximum wind speed threshold? If not, why?
What is the mean windspeed at 140m at these buoy locations? Is the jet windspeed maximum generally higher than the mean windspeeds? I think in order to help make this LLJ identification

algorithm more relevant to other offshore locations, it would be helpful to explain your specific choices and thresholds in a little more detail.

**Minor Comments:**

Line 16: You mention that these cold season LLJs generally occur below 250m and that the warm season LLJs are generally higher. It might be nice to mention the height the warm season LLJs are observed at for context (e.g. higher altitude California coastal jet (typically at heights of 300–400 m) influenced by the North Pacific High).

Line 18: I think you could cut out a couple words in this sentence to simplify it (it is a long sentence). "The lidar buoy observations  support the validation of LLJ representation in atmospheric models  essential for assessing the potential yield of offshore wind farms"

Line 25: It might be helpful for the reader to briefly define the term "false alarm" here.

Lines 38, 145-155: Another study you could take a look at is McCabe and Freedman 2023 (https://doi.org/10.1175/WAF-D-22-0119.1). That paper discusses another shear based approach to identifying LLJs that occur during sea breeze events on the east coast.

Lines 40-45: You mention the impact of the LLJ on wake effects and turbine fatigue, but you don't mention how the LLJ may be able to increase wind speeds across the rotor plane, therefore increasing potential energy production until the end of the paper. I think it might be helpful to add a sentence on that here.

Line 101: Just to clarify, here you mention the buoy being equipped with the Leosphere WindCube 866 lidars, and in the past section (line 81), you mention the AXYS WindSentinel buoys are have Leosphere WindCube v2 lidar systems. Are these lidars the same?

Lines 183-184: You refer to figure 3a and 3b, but figure 3 only has one panel.

Lines 190-194: The second half of this sentence is a little confusing for the reader. I would suggest re-phrasing the sentence (especially the part after "directions associated with ... ") for clarity.

Line 226: I would consider reminding the reader that $L$ (Obukhov length) is defined in equation 1.

Lines 288-289: For ease of reading, it might help to break this sentence into two sentences. You could simply add a period after the word "phenomena".

Line 325 (Figure 10): I think you need some sort of legend, or just an explanation in the figure caption, that clarifies what the numbers on the plots represent.

Lines 340-345: How does the LLJ core wind speed bias vary with height? Is it more pronounced at certain LLJ core heights than others?

Line 378: I would suggest putting the abbreviations you are using as the subscript in equations 2 and 3 for lowest (*lo*) and highest (*hi*) in parenthesis.

---

## Referee Comment (RC3)

**Summary**

The authors present novel offshore LLJ research off the coast of California in regions relevant to wind energy. This research includes not only observational LLJ characterization, but also various model comparisons. Overall, observations capture LLJ events occurring only a few percent over the observational campaign but note that the vertical extent of these observations will undercount LLJ events. The models tend to underestimate the number of LLJ events but overestimate the duration. Models do a good job of capturing the hourly timing, but struggle on a seasonal basis. Other LLJ characteristics have positive/negative biases depending on the location and model used.

Overall, I believe this research to be a great addition to the offshore wind energy literature, and I recommend this article to be *accepted with minor revisions*, which are listed below.

**Major comments**

- I feel the paper ends abruptly and that no proper conclusion is given. It would be beneficial for the paper to include, at the very least, future directions this research could/should go to help offshore low-level jet characterization.

**Minor comments**

- Line 45: It is stated that the California coastal LLJ is well-studied, yet only two references are included. The first reference, Parish 2000, is repeatedly mentioned and referenced from this point on. To the authors knowledge, are these the only two studies to investigate LLJs along the California coast? If so, I would recommend rewording this sentence to tamper down the claim that the California coastal LLJ is well-studied.
- Lines 88-89: Do the phrases "in 1100 m of water" and "in 625 m of water" refer to the depth of the ocean floor to the sea-level surface? This reads weird to me.
- Section 2.1: What is the temporal resolution of the lidars and ultrasonic anemometers, and was any temporal averaging done? I understand the specifics are in Severy et al. (2021) and Krishnamurthy et al. (2023), but I feel at the very least basic temporal resolution of these instruments should be discussed.
- Section 2.2: What is the advantage of using the TOGA COARE algorithm, compared, say, to alternative algorithms? I feel a little more discussion is warranted here.
- Section 2.4: While I am aware that quantifying LLJs are an unsettled topic of discussion in our field, I would be remiss not to request Debnath et al. (2021) to be included in this section. While Debnath et al. (2021) leverages Kalverla et al. (2019)'s work, the Debnath paper has been used extensively in LLJ discussions in the U.S.
- Lines 163-164: I appreciate the mention of the limitation of the observations in capturing the true extent of LLJ events.

- Figure 2b: It is interesting that the longest LLJ occurrence at Morro Bay occurs at the end of the period. I assume this is by chance, no?
- Line 242: When it is said that the data are resampled at the top of the hour, does this mean that only top of the hour observations are taken, or some mathematical operation is done (hourly average)?
- Figure 11: Neat way to visually compare LLJ core height differences!
- Lines 352-353: Is that duration time correct? 10:00 to 14:10 UTC is not 14.2 hours. Am I missing something here?
- Line 513: The paper for this reference has been officially published and should be updated accordingly.

**References**

(Debnath et al., 2021)

Debnath, M., Doubrawa, P., Optis, M., Hawbecker, P., and Bodini, N.: Extreme wind shear events in US offshore wind energy areas and the role of induced stratification, Wind Energy Science, 6, 1043–1059, https://doi.org/10.5194/wes-6-1043-2021, 2021.

---

## Author Comment (AC1)

This study presents a nice comparison of observations from a temporary LiDAR buoy deployment off of the California coast, to the NREL 20-year Wind Resource Dataset (CA20-Ext), the 2023 National Offshore Wind Dataset (NOW-23), and the ERA5 reanalysis product. This work is important as we look forward to offshore wind development on the West Coast, as model errors in low-level jet (LLJ) representation can affect wind turbine energy generation. This study successful compares three different models to offshore lidar observations and identifies the strengths and weaknesses of each.

I do have a few minor comments that I believe will help the clarity and flow of the paper. Some general comments are that I do think that the figures could benefit from panel titles that have the lidar buoy location/model name. Even though it is explained in the figure caption, I think it would be nice to also add titles for quick referencing. I also noticed that the word "respectively" was very used often in the analysis sections (specifically sections 4.1 and 4.2), as well as a few long run-on sentences which made the paper more difficult to read. The rest of my comments are more specific and are listed below.

Overall, I think this is an excellent study and I am excited to see the results of another offshore lidar deployment. It is difficult to get profiler measurements offshore, and the field needs studies like this one, that can showcase the discrepancies between model predictions and observational data for wind energy applications. I really appreciated how the authors compared how the model bias in LLJ prediction could impact wind energy generation forecasts. I recommend this paper goes to publication.

Thank you very much for your review and for your support of our work! We have improved the sentence structure throughout the text and have removed many of the overused instances of "respectively" in sections 4.1 and 4.2. Additionally, we have added information about the buoy locations and models to the figures in axis labels and text boxes. We have addressed your specific suggestions as follows.

**Major Comments:**

Lines 30-35: This paper is missing a figure that shows entire study region, including the location of the 2 lidar buoys (at Morro Bay and Humboldt) as well as the proposed 5 lease areas. I think this would be very helpful in providing some context for those that aren't familiar with the study area.

Thank you for this helpful suggestion. The following map showing the locations of the lidar buoys and the lease areas has been added to the manuscript as Figure 1.

[Figure]

Figure 1. (a) Map of locations of the California DOE lidar buoy deployments and the California wind energy lease areas as of January 2024. (b) Photo of the DOE lidar buoys by Ocean Tech Services, LLC and Pacific Northwest National Laboratory.

Lines 85-90, 188: Again, referencing a map of the study region here would help your discussion.

References to the new Figure 1 have been added at the recommended lines (99 and 223).

Lines 153-156: Why did you choose a 2 ms$^{-1}$ windspeed drop off threshold in your LLJ identification? Is this based off a percentage of the mean wind speed that is typically seen in the region? I think explaining why the 2 ms$^{-1}$ works well in this offshore environment will be helpful in making this study more applicable in other areas.

We felt that 2 m s$^{-1}$ was an appropriate drop off threshold given the vertical extent of the wind profiles that we had to work with (240 m for the observations and 200 m for the model comparisons). Nunalee and Basu (2013) employ a drop off threshold of 4 m s$^{-1}$ above and below the jet core using observations up to 2 km above ground level. Similarly, Carroll et al. (2019) utilise a fall-off threshold of 5 m s$^{-1}$ using wind profiles over the lowest 1.25 km of the atmosphere. We desire to be consistent with the multiple LLJ studies we came across that use wind profiles covering the lowest 500 m or less of the atmosphere, such as Kalverla et al. (2019), Hallgren et al. (2020), and Aird et al. (2022), all of whom employ a drop off threshold of 2 m s$^{-1}$. We have added Lines 184-186 to provide clarity on our decision: "To provide consistency with similar LLJ studies that utilise wind profiles within the lowest hundreds of meters of the atmosphere, this work employs a fall-off threshold of 2 m s$^{-1}$ above and below the core speed to define an LLJ."

Aird, J. A., Barthelmie, R. J., Shepherd, T. J., and Pryor, S. C.: Occurrence of Low-Level Jets over the Eastern U.S. Coastal Zone at Heights Relevant to Wind Energy, Energies, 15(2), 445, https://doi.org/10.3390/en15020445, 2022.

Hallgren, C., Aird, J. A., Ivanfell, S., Körnich, H., Barthelmie, R. J., Pryor, S. C., and Sahlée, E.: Brief communication: On the definition of the low-level jet, Wind Energy Science Discussions [preprint], https://doi.org/10.5194/wes-2023-74, in review, 2023.

Kalverla, P. C., Duncan Jr, J. B., Steeneveld, G-J., and Holstag, A. A. M.: Low-level jets over the North Sea based on ERA5 and observations: together they do better, Wind Energy Science, 4, 2, 193-209, https://doi.org/10.5194/wes-4-193-2019, 2019.

Lines 159-165: Did you consider any other variables when identifying the LLJs? You mention that there is no restriction set on the vertical distance of the 2ms$^{-1}$ fall off, but did you set a minimum or maximum wind speed threshold? If not, why? What is the mean windspeed at 140m at these buoy locations? Is the jet windspeed maximum generally higher than the mean windspeeds? I think in order to help make this LLJ identification algorithm more relevant to other offshore locations, it would be helpful to explain your specific choices and thresholds in a little more detail.

We explored using minimum (8 m s$^{-1}$) and maximum (40 m s$^{-1}$) wind speed thresholds during our initial investigations using these datasets. Ultimately, however, we were curious as to what the observations showed for the simple LLJ definition as a local wind maximum, employing a 2 m s$^{-1}$ fall-off threshold. We found that, using this definition, the ranges of jet core wind speeds at each deployment location were relevant to the cubic portion, maximum power production portion, and the cut-out or derating portion of wind turbine power curves. Lines 234-235 relate the LLJ core speeds to power curves.

The jet wind speed maximum is typically higher than the mean deployment-wide wind speeds at most heights, except the very lowest heights. Per your helpful question on this topic, we added the median deployment-wide wind speeds at each height a.s.l. to Figure 7. We selected the median deployment-wide wind speeds to correspond with the LLJ statistics presented in the box plots. The following discussion was also added to Lines 236-240: "At most measurement heights, the median LLJ core speeds exceed the median deployment-wide wind speeds. The differences between the median LLJ core and deployment-wide wind speeds are especially pronounced at Morro Bay (e.g., differences exceeding 10 m s$^{-1}$ at heights of 200 m and above), where the wind profiles tend to have less shear than those from the Humboldt deployment (Sheridan et al., 2022)."

**Minor Comments:**

Line 16: You mention that these cold season LLJs generally occur below 250m and that the warm season LLJs are generally higher. It might be nice to mention the height the warm season LLJs are observed at for context (e.g. higher altitude California coastal jet (typically at heights of 300–400 m) influenced by the North Pacific High).

Thank you for the helpful suggestion. We have reworded the sentence as follows: "LLJs were observed more frequently in colder seasons within the lowest 250 m above sea level, in contrast with the summertime occurrence of the higher altitude California coastal jet influenced by the North Pacific High which typically occurs at heights of 300 m – 400 m." (Lines 15-17)

Line 18: I think you could cut out a couple words in this sentence to simplify it (it is a long sentence). "The lidar buoy observations also support the validation of LLJ representation in atmospheric models that are essential for assessing the potential yield of offshore wind farms"

We have reduced the sentence to: "The lidar buoy observations also validate LLJ representation in atmospheric models that estimate potential energy yield of offshore wind farms." (Lines 18-19)

Line 25: It might be helpful for the reader to briefly define the term "false alarm" here.

This is a great suggestion for clarity, and we have modified the sentence to read: "However, CA20-Ext also produced the most LLJ false alarms, instances when a model identified an LLJ but no LLJ was observed." (Lines 24-25)

Lines 38, 145-155: Another study you could take a look at is McCabe and Freedman 2023 (https://doi.org/10.1175/WAF-D-22-0119.1). That paper discusses another shear based approach to identifying LLJs that occur during sea breeze events on the east coast.

Thank you for this important reference. We have added McCabe and Freedman (2023) to the list of sources that provide analyses on meteorological impacts on wind profiles on Line 41. Additionally, we have added the following to the discussion of LLJ identification: "McCabe and Freedman (2023) define an LLJ as having a shear exponent at least ±0.2 above and below the jet maximum, along with a wind speed threshold based on the mean wind speeds in the rotor plane during days with sea breeze events." (Lines 182-184)

Lines 40-45: You mention the impact of the LLJ on wake effects and turbine fatigue, but you don't mention how the LLJ may be able to increase wind speeds across the rotor plane, therefore increasing potential energy production until the end of the paper. I think it might be helpful to add a sentence on that here.

Thank you for pointing out this omission. We have added the following to the discussion of LLJ impacts: "LLJs can result in significant acceleration of the wind speed at heights within the wind turbine rotor-swept area (Banta et al., 2008), which can lead to increases in wind energy generation." (Lines 43-44)

Banta, R. M., Pichugina, Y. L., Kelley, N. D., Jonkman, B., and Brewer, W. A.: Doppler lidar measurements of the Great Plains low-level jet: Applications to wind energy, IOP Conference Series: Earth and Environmental Science, 1, 012020, doi:10.1088/1755-1307/1/1/012020, 2008.

Line 101: Just to clarify, here you mention the buoy being equipped with the Leosphere WindCube 866 lidars, and in the past section (line 81), you mention the AXYS WindSentinel buoys are have Leosphere WindCube v2 lidar systems. Are these lidars the same?

They are the same lidar systems and we appreciate you mentioning how this can be confusing. We have changed the mention of "v2" to "866" for consistency. Thank you!

Lines 183-184: You refer to figure 3a and 3b, but figure 3 only has one panel.

Thank you for pointing out this typo! We have changed the references of Figure 3a and 3b to simply Figure 5 (based on the updated figure numbering from new figures suggested by the reviewers).

Lines 190-194: The second half of this sentence is a little confusing for the reader. I would suggest re-phrasing the sentence (especially the part after "directions associated with ... ") for clarity.

We agree, and have reorganized the sentence as follows: "During LLJ events, the rotor layer winds at Humboldt tend to be land-based, from the north-northeast (0°–20°) and Cape Mendocino to the south-southeast (160°-180°), though some LLJ events are associated with the offshore flow directions of 180°-220° (Figure 6c)." (Lines 224-226)

Line 226: I would consider reminding the reader that $L$ (Obukhov length) is defined in equation 1.

Good idea. The reminder to reference Eq. 1 has been added to that sentence as follows. "The deployment-wide assessments of Sheridan et al. (2022) found predominantly near-neutral atmospheric stability ($z/L \approx 0$) at $z = 4$ m a.s.l. and L as defined in Eq. 1 for both Humboldt and Morro Bay." (Line 266)

Lines 288-289: For ease of reading, it might help to break this sentence into two sentences. You could simply add a period after the word "phenomena".

Thank you for the suggestion and the solution. The sentence has been split into two sentences per your recommendation. (Lines 335-337)

Line 325 (Figure 10): I think you need some sort of legend, or just an explanation in the figure caption, that clarifies what the numbers on the plots represent.

We agree and have added the following to the caption: "The markers are coloured according to season and are labelled by month (1 = January, 2 = February, ... 12 = December)."

Lines 340-345: How does the LLJ core wind speed bias vary with height? Is it more pronounced at certain LLJ core heights than others?

We appreciate the suggestion to add an analysis of the LLJ core wind speed bias according to height, and have replaced the scatter plot in Figure 11 (now Figure 14) with just such an analysis. We found that overestimation of the observed LLJ core wind speed occurs at the lowest observed LLJ core heights, with a trend of biases approaching or achieving underestimation with increasing observed core height, and have added this discussion to Lines 294-396.

Line 378: I would suggest putting the abbreviations you are using as the subscript in equations 2 and 3 for lowest (*lo*) and highest (*hi*) in parenthesis.

We have added (*lo*) and (*hi*) to the sentence as you suggest.

---

## Author Comment (AC2)

Review of Sheridan et al. 2023: Offshore low-level jet observations and model representation using lidar buoy data off the California coast

First of all, I want to congratulate you on a very nice piece of work, that is really well written and is an important scientific contribution to the wind energy community. The figures are clear and easy to interpret and new creative ways of presenting the results are used, providing great insights in the data. It was a joy reading a manuscript that seems ready for publication as is.

The authors study observations of the wind profiles at two sites off the California coast, assessing them for low-level jets (LLJs). The seasonality and diurnal cycle of the appearance of the LLJs is investigated as well as key characteristics such as LLJ core height, core speed, and duration of LLJ events. Further, the authors validate three different models (the global ERA5 reanalysis and two regional models with different planetary boundary layer schemes) in terms of accurately resolving LLJs.

I only have some minor comments that I, humbly, think can enhance the manuscript further:

Thank you very much for the kind support of our work! We appreciated all of your suggestions for improvement and have incorporated them as follows.

L35: It would be helpful for the reader to add a map showing the exact locations of Humboldt and Morro Bay, including names of sites (such as Cape Mendocino and Point Sur that you mention in the text) on the map. Perhaps also photos of the buoys and a visual presentation of the data availability from Humboldt and Morro Bay could be included in this Figure.

Thank you for this helpful suggestion. The below map showing the locations of the lidar buoys, the wind energy lease areas, and the geographic features including Cape Mendocino and Point Sur has been added to the manuscript as Figure 1, along with a photo of the lidar buoys.

[Figure]

Figure 1. (a) Map of locations of the California DOE lidar buoy deployments and the California wind energy lease areas as of January 2024. (b) Photo of the DOE lidar buoys by Ocean Tech Services, LLC and Pacific Northwest National Laboratory.

We also appreciate the idea of visual representation of data availability and have added Figure 2 to the manuscript to address this recommendation.

[Figure]

Figure 2. Availability of the near surface and lidar wind data at (a) Humboldt and (b) Morro Bay during the study period.

L47-48: 300 – 400 m is still in a layer above the top of the rotor of an offshore wind turbine. It would be good to give an example of the size (hub height and rotor diameter) of a modern offshore wind turbine, such as the NREL 15 MW turbine you refer to later on.

Great idea! We have used the Offshore Wind Market Report: 2023 Edition to provide real-world comparisons of offshore wind turbine dimensions with the summertime LLJ core heights as follows:

"The jet occurs at the top of the marine boundary layer with frequent core heights of 300 m – 400 m (Parish 2000), which are above the 2022 average offshore wind turbine hub height (116.6 m) and rotor diameter (174.6 m) (Musial et al., 2023). However, Musial et al. (2023) reported significantly larger hub heights (near 160 m) and rotor diameters (near 250 m) from future project announcements, leading to prospective rotor-swept upper limits approaching summertime LLJ core heights." (Lines 52-56)

Musial, W., Spitsen, P., Beiter, P., Duffy, P., Mulas Hernando, D., Hammond, R., Shields, M., and Marquis, M.: Offshore Wind Market Report: 2023 Edition, Tech. Rep., U.S. Department of Energy Office of Energy Efficiency Renewable Energy, https://doi.org/10.2172/1997466, 2023.

L54-55: The New European Wind Atlas (e.g., Dörenkämper et al. 2020) would also be a nice reference here

The reference to the New European Wind Atlas has been added to the discussion of wind resource products that use ERA5 as an input boundary condition per your suggestion. (Line 63)

L66: To add more information on the offshore wind profile and LLJs and how they are affected by the selection of the PBL scheme, I would suggest adding a couple of sentences summarizing the results from Svensson et al. (2016)

Thank you for recommending this important paper for the literature review. We have added the following to the manuscript, along with the reference you provided: "Svensson et al. (2016) found that no single PBL scheme among six evaluated in WRF outperformed the others in representing the wind speed and temperature profiles, the jet core height and wind speed, and the maximum wind shear during three LLJ case studies over the Baltic Sea." (Lines 75-77)

L94: Here it could just be clarified that: 17 October 2020 to 30 September 2021 *(i.e., almost a year)* for Morro Bay and 17 October 2020 to 27 December 2020 and 24 May 2021 to 30 September 2021 *(i.e., six months in total)* for Humboldt.

We appreciate the helpful suggestion and have modified the sentence per your recommendation: "The final periods of record employed in this study are 17 October 2020 to 30 September 2021 (i.e., almost a year) for Morro Bay and 17 October 2020 to 27

December 2020 and 24 May 2021 to 30 September 2021 (i.e., almost seven months) for Humboldt..." (Lines 105-107)

L106: Clarification: intervals of 20 m between 40 m and 240 m a.s.l. *(i.e., 11 height levels)*

"(i.e., 11 height levels)" has been added to this line per your helpful suggestion. (Line 124)

L108: Here the information that data was collected as 10 min averages should be added. Also, a few words on the quality control of the data would be nice.

The following sentences concerning the observations have been added to Section 2.1 per your recommendation: "The temporal resolutions of the lidar and near-surface measurements utilised in this analysis are 10-minute averages. Quality control of the measurements, discussed in detail in Krishnamurthy et al. (2023) included making sure sensors were not reporting beyond the manufacturer limits, comparisons with nearby sensors, removal of abnormal spikes in the data, physics-based analyses, motion correction, and filtering based on the signal-to-noise ratio for the lidars." (Lines 120-123)

Krishnamurthy, R., García Medina, G., Gaudet, B., Gustafson Jr, W. I., Kassianov, E., Liu, J., Newsom, R. K., Sheridan, L., and Mahon, A.: Year-long Buoy-Based Observations of the Air-Sea Transition Zone off the U.S. West Coast, Earth System Science Data, 15(12), 5667-5699, https://doi.org/10.5194/essd-15-5667-2023, 2023.

L159 and throughout the manuscript: Using comma signs as thousand separators could increase the readability of the large numbers (such as 24,878 observations on L159)

Commas have been added to large numbers throughout the text.

L170-171: As the distributions of LLJ event duration is highly skewed (as also shown in Figure 8), the median duration should also be presented, accompanying the mean value.

Good idea. We have updated the sentences accordingly:

"The average (median) LLJ duration at Humboldt was 1.6 hours (0.8 hours)." (Line 202)

"The average (median) LLJ duration at Morro Bay was 2.2 hours (1.0 hour)." (Line 203)

L183-184: You refer to Figure 3a and Figure 3b, but there are no panels in Figure 3.

Thank you for pointing out this typo! We have changed the references of Figure 3a and 3b to simply Figure 5 (based on the updated figure numbering from new figures suggested by the reviewers).

L185: Commenting on the diurnal cycle presented in Figure 3, I think you should also mention that the comparison between the two sites is possibly biased because of the data

outage at Humboldt. It would be great if you could add confidence intervals around the lines in Figure 3 to motivate how certain you can be if there actually is a diurnal cycle in LLJ occurrence.

Per your helpful suggestion, we have reworded the discussion on the diurnal cycle of LLJs as follows: "At Morro Bay, a dependency on the time of day is noted for LLJ presence, with more LLJs occurring in the hours before local midnight and fewer LLJs occurring during the morning, afternoon, and early evening (Figure 5). At Humboldt, however, LLJs occurred across the diurnal cycle with little variation in frequency (Figure 5), though the comparison may be biased due to the data outage during the Humboldt deployment." (Lines 216-219)

Additionally, we have added the 95% confidence intervals to the figure:

[Figure]

Figure 5. Diurnal trends in LLJ occurrences during the Humboldt and Morro Bay lidar buoy deployments with 95% confidence intervals.

Figure 6: In the caption, if you could write "bulk wind shear below the jet core", that would remind the reader and help interpreting the plot.

"bulk wind shear below the jet core" has been added to the caption of the new Figure 8 per your nice suggestion.

L242: Is it the closest grid points that have been used? Please specify the distance between the grid points used and Humboldt and Morro Bay, respectively.

The closest grid points were used and the following text has been added to provide clarification: "Horizontally, the nearest model grid point to each buoy is selected for

evaluation. The Humboldt buoy location is 8.3 km from the nearest ERA5 grid point and 1.1 km from the nearest CA20-Ext and NOW-23 grid points. The Morro Bay buoy location is 10.7 km from the nearest ERA5 grid point and 0.7 km from the nearest CA20-Ext and NOW-23 grid points." (Lines 284-286)

L247: Also, Kalverla et al. (2020) should be cited here (see Figure 6 in that paper)

The citation has been added as follows, thank you. "To evaluate the performance of models for LLJ representation off the California coast, we utilize the following methodology from Hallgren et al. (2020) and Kalverla et al. (2020) to categorize whether each model captures, misses, or incorrectly reports an LLJ at a given timestamp." (Lines 288-290)

L252-256: In addition to the success rates, the frequency bias should be included.

Thank you for this helpful suggestion to add a new helpful metric for model performance. We have provided the following text concerning frequency bias:

"While CA20-Ext produced the most LLJ hits for the California deployment, it also produced the most false alarms across the models (150 and 193 at Humboldt and Morro Bay, respectively). The false alarms contribute significantly to the CA23-Ext frequency bias, the ratio of the total number of predicted LLJs to the total number of observed LLJs (Hallgren et al., 2020). A frequency bias of one is a perfect score, while a frequency bias above (below) one indicates model overestimation (underestimation) of the number of observed LLJs. The CA20-Ext frequency biases of 2.5 (Humboldt) and 1.4 (Morro Bay) represent significant model overestimation of observed LLJs, particularly at Humboldt. NOW-23 produced 41 and 70 false alarms at Humboldt and Morro Bay which, combined with the hits, results in frequency biases indicative of model underestimation (0.8 and 0.5, respectively). ERA5 produced 2 and 6 false alarms at Humboldt and Morro Bay, generating frequency biases emblematic of significant model underestimation (<0.1 at both sites)." (Lines 302-310)

Hallgren, C., Arnqvist, J., Ivanell, S., Körnich, H., Vakkari, V., and Sahlée, E.: Looking for an Offshore Low-Level Jet Champion among Recent Reanalyses: A Tight Race over the Baltic Sea, Energies, 13(14), 3670, https://doi.org/10.3390/en13143670, 2020.

Figure 13: As you are commenting on the prior LLJ event in the text, I would love to see the time series starting already at 07:00 UTC to also include this event in the plot. Perhaps you could also mark the LLJ core for each time step where an LLJ is present in the panels?

Per your helpful idea, the figure now begins at 7:00 UTC and the LLJ core heights have been indicated.

[Figure]

Figure 15. (a) Observed, (b) CA20-Ext-simulated, and (c) NOW-23-simulated wind speeds during an observed low-level jet on 15 January 2021 at Morro Bay. Reported timestamps are in UTC. Markers indicate the (a) observed and (b), (c) modelled jet core heights.

L375: Using the rotor equivalent wind speed (REWS) instead of only the hub height wind speed to calculate the estimated power production would really strengthen the work. For reference see e.g., St. Pé et al. (2018) and Murphy et al. (2020).

Excellent suggestion! We have reworked all of the analysis in the Discussion to utilise the REWS instead of just the hub height wind speed.

L409: Finally, adding a paragraph discussing general similarities and differences when comparing your results with LLJ studies from e.g., the US Atlantic coast, the North Sea, and the Baltic Sea, would put your work into a broader context and would be very valuable for future reference.

We agree, and combined your idea with another reviewer's suggestion for the final paragraph as follows: "Coastal and offshore measurement campaigns, while challenging to execute, provide valuable data collections to support the evaluation of potential wind energy generation in an offshore setting. The increasing number of such deployments is advantageous for understanding the characteristics of meteorological influences, such as LLJs, on the wind profile in unique locations. For example, recent measurement campaigns yielded locationally-driven diversity in the time of year for most frequent LLJ occurrence, namely May in the Baltic Sea (Hallgren et al., 2020), April – November in the New York Bight (McCabe and Freedman, 2023), and January at Morro Bay. Offshore observations are also needed for highlighting research areas for wind modelling improvement, such as the studies of Hallgren et al. (2020) and this work in noting the limitations of ERA5 representation of LLJs in distinct environments. The breadth of wind profile characteristics revealed by such measurement campaigns encourages similar analyses in new areas of offshore wind development interest. Subsequent DOE lidar buoy deployments include the waters off Hawaii and the U.S. Atlantic coast. Additionally, we look forward to expansion in the understanding of offshore LLJ occurrence and features, particularly in vertical extent, as floating lidar technology continues to advance. We hope this work encourages increased offshore wind observational campaigns to support validation and improvements for modelling of atmospheric phenomena like LLJs." (Lines 459-471)

References:

Dörenkämper , Martin, et al. "The Making of the New European Wind Atlas – Part 2: Production and Evaluation." Geoscientific Model Development 13.10 (2020): 5079-5102, doi: 10.5194/gmd-13-5079-2020)

Kalverla, Peter C., et al. "Quality of wind characteristics in recent wind atlases over the North Sea." Quarterly Journal of the Royal Meteorological Society 146.728 (2020): 1498-1515, doi: 10.1002/qj.3748

Murphy, Patrick, Julie K. Lundquist, and Paul Fleming. "How wind speed shear and directional veer affect the power production of a megawatt-scale operational wind turbine." Wind Energy Science 5.3 (2020): 1169-1190, doi: 10.5194/wes-5-1169-2020

St. Pé, Alexandra, et al. "Classifying rotor-layer wind to reduce offshore available power uncertainty." Wind Energy 21.7 (2018): 461-473, doi: 10.1002/we.2159

Svensson, Nina, et al. "Stable atmospheric conditions over the Baltic Sea: model evaluation and climatology." Boreal Environment Research, ISSN 1239-6095, E-ISSN 1797-2469, Vol. 21, s. 387-404 (2016)

---

## Author Comment (AC3)

The authors present novel offshore LLJ research off the coast of California in regions relevant to wind energy. This research includes not only observational LLJ characterization, but also various model comparisons. Overall, observations capture LLJ events occurring only a few percent over the observational campaign but note that the vertical extent of these observations will undercount LLJ events. The models tend to underestimate the number of LLJ events but overestimate the duration. Models do a good job of capturing the hourly timing, but struggle on a seasonal basis. Other LLJ characteristics have positive/negative biases depending on the location and model used.

Overall, I believe this research to be a great addition to the offshore wind energy literature, and I recommend this article to be *accepted with minor revisions*, which are listed below.

We are so grateful for your support of our work and for the helpful suggestions for improvement! We have incorporated your recommendations as follows.

Major comments

- I feel the paper ends abruptly and that no proper conclusion is given. It would be beneficial for the paper to include, at the very least, future directions this research could/should go to help offshore low-level jet characterization.

  We agree that the paper ended on a lackluster note. We have revised the final paragraph as follows per your and another reviewer's helpful recommendations: "Coastal and offshore measurement campaigns, while challenging to execute, provide valuable data collections to support the evaluation of potential wind energy generation in an offshore setting. The increasing number of such deployments is advantageous for understanding the characteristics of meteorological influences, such as LLJs, on the wind profile in unique locations. For example, recent measurement campaigns yielded locationally-driven diversity in the time of year for most frequent LLJ occurrence, namely May in the Baltic Sea (Hallgren et al., 2020), April – November in the New York Bight (McCabe and Freedman, 2023), and January at Morro Bay. Offshore observations are also needed for highlighting research areas for wind modelling improvement, such as the studies of Hallgren et al. (2020) and this work in noting the limitations of ERA5 representation of LLJs in distinct environments. The breadth of wind profile characteristics revealed by such measurement campaigns encourages similar analyses in new areas of offshore wind development interest. Subsequent DOE lidar buoy deployments include the waters off Hawaii and the U.S. Atlantic coast. Additionally, we look forward to expansion in the understanding of offshore LLJ occurrence and features, particularly in vertical extent, as floating lidar technology continues to advance. We hope this work encourages increased offshore wind observational campaigns to support validation and improvements for modelling of atmospheric phenomena like LLJs." (Lines 459-471)

Minor comments

- Line 45: It is stated that the California coastal LLJ is well-studied, yet only two references are included. The first reference, Parish 2000, is repeatedly mentioned and referenced from this point on. To the authors knowledge, are these the only two studies to investigate LLJs along the California coast? If so, I would recommend rewording this sentence to tamper down the claim that the California coastal LLJ is well-studied.

  Thank you for pointing this out. We were remiss in capturing the literature on the California summertime LLJ and have provided a more complete set of citations to the sentence: "The summertime California coastal LLJ is well-studied and occurs due to the pressure gradient between the North Pacific High and southwestern U.S. thermal low (Burk and Thompson, 1996; Holt, 1996; Parish 2000; Pomeroy and Parish, 2001; Ström and Tjernström, 2004; Liu et al., 2023)." (Lines 50-52)

  Burk, S. D. and Thompson, W. T.: The Summertime Low-Level Jet and Marine Boundary Layer Structure along the California Coast, Monthly Weather Review, 124(4), 668-686, https://doi.org/10.1175/1520-0493(1996)124%3C0668:TSLLJA%3E2.0.CO;2, 1996.

  Holt, T. R.: Mesoscale forcing of a boundary layer jet along the California coast, Journal of Geophysical Research: Atmospheres, 101(D2), 4235-4254, https://doi.org/10.1029/95JD03231, 1996.

  Liu, Y., Gaudet, B., Krishnamurthy, R., Tai, S. L., Berg, L. K., Bodini, N., and Rybchuk, A.: Identifying meteorological drivers for errors in modelled winds along the Northern California Coast, Monthly Weather Review, https://doi.org/10.1175/MWR-D-23-0030.1, 2023.

  Parish, T. R.: Forcing of the Summertime Low-Level Jet along the California Coast, Journal of Applied Meteorology and Climatology, 39(12) 2421-2433, https://doi.org/10.1175/1520-0450(2000)039%3C2421:FOTSLL%3E2.0.CO;2, 2000.

  Pomeroy, K. R. and Parish, T. R.: A Case Study of the Interaction of the Summertime Coastal Jet with the California Topography, Monthly Weather Review, 129(3), 530-539, https://doi.org/10.1175/1520-0493(2001)129%3C0530:ACSOTI%3E2.0.CO;2, 2001.

  Ström, L. and Tjernström, M.: Variability in the summertime coastal marine atmospheric boundary-layer off California, USA, Quarterly Journal of the Royal Meteorological Society, 130(597), 423-448, https://doi.org/10.1256/qj.03.12, 2004.

- Lines 88-89: Do the phrases "in 1100 m of water" and "in 625 m of water" refer to the depth of the ocean floor to the sea-level surface? This reads weird to me.

  Thank you for this suggestion to improve the clarity. We have switched the wording to "in a depth of 1100 m of water" and "in a depth of 625 m of water." (Lines 99-102)

- Section 2.1: What is the temporal resolution of the lidars and ultrasonic anemometers, and was any temporal averaging done? I understand the specifics are in Severy et al. (2021) and Krishnamurthy et al. (2023), but I feel at the very least basic temporal resolution of these instruments should be discussed.

  We agree. The temporal resolution in the original draft was not mentioned until the results section, and it is far more helpful to add this information to the data discussion. We have added the following sentence to Section 2.1: "The temporal resolutions of the lidar and near-surface measurements utilised in this analysis are 10-minute averages." (Line 120)

  Additionally, the discussion of ERA5 in Section 2.3 was supplemented with temporal information as follows: "ERA5 provides extensive temporal coverage from 1950 through present time at 1-hour temporal resolution (Hersbach et al., 2020)." (Line 163)

  The discussion of CA20-Ext and NOW-23 in Section 2.3 had already made note of their temporal resolutions in the original draft, so this wording remains unchanged: "CA20-Ext and NOW-23 include 61 vertical layers and, as CA20 extensions to evaluate the PBL schemes, output wind estimates at 10 m and every 20 m between 20 m and 200 m a.s.l. at 5-minute temporal resolution and 2-km horizontal resolution." (Lines 156-158)

- Section 2.2: What is the advantage of using the TOGA COARE algorithm, compared, say, to alternative algorithms? I feel a little more discussion is warranted here.

  Thank you for this suggestion. We have added the following discussion to Section 2.2: "To compute the traditional measure of surface layer stability, $z/L$, heat and momentum turbulent fluxes are needed, but measurements of these fluxes are not available to us. The developers of the COARE series of parameterizations provide iterative algorithms that relate these fluxes to measured mean state thermodynamic and wind fields in a self-consistent manner that is also consistent with Monin-Obukhov similarity theory. The COARE parameterizations are specifically adapted to the ocean environment, for which the wave state must either be provided or parameterized as a function of wind speed so that the turbulent momentum flux may be determined. Multiple marine observational datasets of momentum flux have been used by the COARE developers over the years to determine and refine these relationships for general global applications." (Lines 136-143)

- Section 2.4: While I am aware that quantifying LLJs are an unsettled topic of discussion in our field, I would be remiss not to request Debnath et al. (2021) to be included in this section. While Debnath et al. (2021) leverages Kalverla et al.

(2019)'s work, the Debnath paper has been used extensively in LLJ discussions in the U.S.

Thank you! The Debnath paper has been added to Section 2.4 as follows: "Debnath et al. (2021) consider a wind profile to be an LLJ if the wind speed gradient between the bottom of the profile and the jet core exceeds a threshold shear value of 0.035 s$^{-1}$ and the wind speed fall-off above the core is at least 1.5 m s$^{-1}$ and 10% of the core speed." (Lines 180-182)

- Lines 163-164: I appreciate the mention of the limitation of the observations in capturing the true extent of LLJ events.

  Thank you!

- Figure 2b: It is interesting that the longest LLJ occurrence at Morro Bay occurs at the end of the period. I assume this is by chance, no?

  We are so glad you pointed this out, because while the longest LLJ at Morro Bay is real (see the heatmap below), the temporal extent of Figure 2b (now Figure 4b) ends prematurely (see annotated graphic below), making it seem like this LLJ is occurring at the end of the analysis period. The long LLJ begins on 20 September 2021 11:40 and ends 21 September 2021 10:00. Also shown in the heatmap is a shorter duration LLJ ranging from 20 September 2021 8:20 to 10:20. We realized that the analysis had truncated the last LLJ in each timeseries and have fixed the graphic so that the temporal extent reaches the end of the analysis period on 30 September 2021, on which another, shorter LLJ occurred which was cut off in the original figure and analysis. The truncation of the last LLJ in the duration analyses was also corrected in the model comparison analysis in Section 4.1. Thank you!

[Figure]

Original Figure 2. Duration in hours of observed LLJs during the (a) Humboldt and (b) Morro Bay lidar buoy deployments.

[Figure]

Figure 4. Duration in hours of observed LLJs during the (a) Humboldt and (b) Morro Bay lidar buoy deployments.

- Line 242: When it is said that the data are resampled at the top of the hour, does this mean that only top of the hour observations are taken, or some mathematical operation is done (hourly average)?

  Only the top of the hour observations are taken. The sentence has been modified as follows to provide clarity: "In order to compare the performance of LLJ representation in wind models, the lidar buoy observations and model wind data are resampled to include only the top of the hour output to temporally align with the ERA5, which has the coarsest temporal resolution (hourly)." (Lines 280-282)

- Figure 11: Neat way to visually compare LLJ core height differences!

  Thank you!

- Lines 352-353: Is that duration time correct? 10:00 to 14:10 UTC is not 14.2 hours. Am I missing something here?

  Thank you for catching this typo! The sentence has been modified to read 4.2 hours instead of 14.2 hours. (Line 404)

- Line 513: The paper for this reference has been officially published and should be updated accordingly.

  Thank you, we have updated the reference with the published DOI.

References

(Debnath et al., 2021)

Debnath, M., Doubrawa, P., Optis, M., Hawbecker, P., and Bodini, N.: Extreme wind shear events in US offshore wind energy areas and the role of induced stratification, Wind Energy Science, 6, 1043–1059, https://doi.org/10.5194/wes-6-1043-2021, 2021.